# Methane emissions from thermokarst lakes must emphasize the ice-melting impact on the Tibetan Plateau

Cuicui Mu [1,2,3] ✉, Pengsi Lei[1], Mei Mu[1], Chunling Zhang[1], Zhensong Zhou[1], Jinyue Song[1], Yunjie Jia[1], Chenyan Fan[1], Xiaoqing Peng[1], Guofei Zhang [1], Yuanhe Yang [4], Lei Wang[5], Dongfeng Li [6], Chunlin Song [7], Genxu Wang[7] & Zhen Zhang[8]

Thermokarst lakes, serving as significant sources of methane ($CH_4$), play a crucial role in affecting the feedback of permafrost carbon cycle to global warming. However, accurately assessing $CH_4$ emissions from these lakes remains challenging due to limited observations during lake ice melting periods. In this study, by integrating field surveys with machine learning modeling, we offer a comprehensive assessment of present and future $CH_4$ emissions from thermokarst lakes on the Tibetan Plateau. Our results reveal that the previously underestimated $CH_4$ release from lake ice bubble and water storage during ice melting periods is $11.2 \pm 1.6$ Gg C of $CH_4$, accounting for $17 \pm 4\%$ of the annual total release from lakes. Despite thermokarst lakes cover only 0.2% of the permafrost area, they annually emit $65.5 \pm 10.0$ Gg C of $CH_4$, which offsets 6.4% of the net carbon sink in alpine grasslands on the plateau. Considering the loss of lake ice, the expansion of thermokarst lakes is projected to lead to 1.1–1.2 folds increase in $CH_4$ emissions by 2100. Our study allows foreseeing future $CH_4$ emissions from the rapid expanding thermokarst lakes and sheds new lights on processes controlling the carbon-climate feedback in alpine permafrost ecosystems.

Permafrost regions, which cover approximately 22% of the Northern Hemisphere land area, store more than 50% of the world's soil organic carbon[1,2]. The onset of global warming is triggering widespread abrupt permafrost thaw[3], which serves as a substantial source of land-atmosphere carbon exchange and has the potential to act as a positive feedback to climate warming[4]. Thermokarst lakes, are characteristic features of abrupt permafrost thaw resulting from the melting of underground ice[5]. These lakes play a crucial role in climate feedback due to their methane ($CH_4$) emission potential compared to surrounding soils, yet they remain the most uncertain source of $CH_4$

[1]Key Laboratory of Western China's Environmental Systems (Ministry of Education), College of Earth and Environmental Sciences, Observation and research station on Eco-Environment of Frozen Ground in the Qilian Mountains, Lanzhou University, Lanzhou, China. [2]State Key Laboratory of Frozen Soil Engineering, Northwest Institute of Eco-Environment and Resources, Chinese Academy of Sciences, Lanzhou, China. [3]Qinghai–Beiluhe Plateau Frozen Soil Engineering Safety National Observation and Research Station, Lanzhou, China. [4]State Key Laboratory of Vegetation and Environmental Change, Institute of Botany, Chinese Academy of Sciences, Beijing, China. [5]Advanced Interdisciplinary Institute of Environment and Ecology, Beijing Normal University, Zhuhai, China. [6]Key Laboratory for Water and Sediment Sciences, Ministry of Education, College of Environmental Sciences and Engineering, Peking University, Beijing, China. [7]State Key Laboratory of Hydraulics and Mountain River Engineering, College of Water Resource and Hydropower, Sichuan University, Chengdu, China. [8]National Tibetan Plateau Data Center, State Key Laboratory of Tibetan Plateau Earth System, Environment and Resource, Institute of Tibetan Plateau Research, Chinese Academy of Sciences, Beijing, China. ✉e-mail: mucc@lzu.edu.cn

emissions in the global permafrost zone[6–8]. Notably, large quantities of $CH_4$ are produced in the anaerobic environments of thermokarst lakes, becoming trapped in lake ice and the water body beneath the ice[9,10]. Upon the melting of lake ice, $CH_4$ undergoes sudden release. However, due to the harsh alpine environment and high monitoring costs, in-situ measurements of lake $CH_4$ release, particularly during ice-covered and ice-melting periods are sparse, posing a significant challenge and hindering our understanding of $CH_4$ release dynamics from thermokarst lakes.

The Tibetan Plateau represents the largest alpine permafrost region globally, covering an area of 1.05 million $km^2$, which accounts for roughly 75% of the total alpine permafrost regions in the Northern Hemisphere[11,12]. This region stores an estimated 15.3–46.2 Pg of organic carbon in the top 3 meters of soil[13]. Similar to the Arctic region, the Tibetan Plateau is experiencing rapid climate warming[14] and substantial permafrost thaw[2]. Presently, it hosts over 160,000 thermokarst lakes, spanning approximately 2825 $km^2$ in total area[15]. The expansion of these lakes is expanding across the plateau. Over the past three decades, the number and area of thermokarst lakes near the Qinghai-Tibet Highway have increased by 59% and 83%, respectively[16]. Changes in lake ice phenology, influencing the duration of ice cover, are undergoing significant alterations by modifying freeze-up and break-up times[17]. Thermokarst lakes emerge as important $CH_4$ sources on the Tibetan Plateau[18,19]. However, uncertainty persists regarding $CH_4$ emissions during ice-melting periods compared to the Arctic region, resulting in the underestimation of current $CH_4$ emissions on the Tibetan Plateau. Additionally, the absence of thermokarst lake simulations for $CH_4$ release in Earth system models stems from a poor understanding of $CH_4$ release dynamics, obscuring projections regarding future changes in $CH_4$ release from thermokarst lakes[3,20]. These knowledge gaps impede the accurate prediction of carbon-climate feedback in alpine permafrost regions under forthcoming climate scenarios.

In this study, we conducted field observations of 56 thermokarst lakes during ice-covered periods (March-April) in 2023 and synthesized our previous measurements of $CH_4$ release from 162 thermokarst lakes during ice-free periods (May-October) on the Tibetan Plateau from 2019 to 2021[18] (Fig. 1a, Supplementary Fig. 1, Supplementary Table 1). In this field survey, we collected the samples of lake ice and water beneath the ice by cutting a hole in the lake ice with a diameter of 30–50 cm (Fig. 1b-e). We measured dissolved $CH_4$ and $CO_2$ concentrations and their stable carbon isotopes to study $CH_4$ production pathways. Subsequently, we illustrated the patterns of $CH_4$ release during both ice-melting (including ice bubble and water storage) and ice-free periods (including ebullition and diffusion) on the whole plateau, thereby estimating the $CH_4$ emissions from thermokarst lakes. Finally, by integrating changes in the susceptibility distribution of thermokarst lake simulated by machine learning models, we predicted the expansion of thermokarst lakes and future changes in $CH_4$ emissions during the ice-melting and ice-free periods under Shared Socioeconomic Pathways (SSPs). We also separately presented future projections of $CH_4$ emissions from thermokarst lakes with and without considering the loss of lake ice.

## Results and Discussion

### $CH_4$ emission pathways

To explore the difference in dissolved $CH_4$ concentrations between ice-free and ice-covered periods on the Tibetan Plateau, we conducted field observations in March-April 2023 and integrated our published data from thermokarst lakes during 2019–2023[18]. The same sampling method was used in two periods in order to narrow the uncertainty (See Methods). Results indicate that during ice-covered period, the median concentration of dissolved $CH_4$ in the water body beneath lake ice is 2.61 µmol/L ($n = 151$; Fig. 1f), with a wide range from 0.03–280.47 µmol/L. Conversely, during ice-free period, the median

dissolved $CH_4$ concentration in lake water is 0.88 µmol/L, ranging from 0.01–39.98 µmol/L ($n = 353$; Fig. 1f). Dissolved $CH_4$ concentrations during ice-covered period significantly surpass those in the ice-free period ($p < 0.001$), with concentrations within the same thermokarst lake reaching up to 100 times higher in the ice-covered period. It is attributable to that methanogenesis continues under the lake ice due to the anaerobic environment and the barrier effect of lake ice[9,21], leading to the accumulation and entrapment of $CH_4$ in ice-covered thermokarst lakes. Interestingly, dissolved $CH_4$ concentrations during ice-covered and ice-free periods exhibit significantly positive correlations (Fig. 1g, Supplementary Table 2).

Additionally, dissolved $CH_4$ concentrations are closely related to sediment organic carbon content in thermokarst lakes with different vegetation types on the Tibetan Plateau (Supplementary Fig. 2). This is attributable to that methanogenesis primarily occurs in the lake sediments, where organic carbon provides crucial substrate for microbial $CH_4$ production[22–24]. The gradient of sediment organic carbon contents in thermokarst lakes is controlled by soil organic carbon around the lakes within the watersheds, which is transported into the thermokarst lakes through hydrological processes[25,26]. Thus, the highest $CH_4$ concentrations are found in thermokarst lakes under the vegetation of alpine swamp meadows (ASM) and meadows (AM), followed by alpine steppe (AS) and desert (AD), evident both ice-covered and ice-free periods (Supplementary Fig. 3). It was shown that sediment organic carbon contents in the thermokarst lakes is closely related to vegetation types on the Tibetan Plateau[18,25]. The results suggest that large amounts of $CH_4$ are trapped in ice-covered thermokarst lakes, exhibiting similar patterns to those during ice-free period.

Furthermore, to elucidate the pathways of $CH_4$ production in thermokarst lakes during ice-covered periods on the Tibetan Plateau, we calculated the carbon fractionation factor ($\alpha_C$) using $\delta^{13}C$ values of dissolved $CO_2$ and $CH_4$. The $\alpha_C$ values greater than 1.055 indicate $CH_4$ origin predominantly via $CO_2$ reduction, while values between 1.040 and 1.055 suggest acetate fermentation[27,28]. The $\alpha_C$ values below 1.040 are likely associated with $CH_4$ oxidation[29] (see Methods). Results show that the observed $\delta^{13}C$-$CH_4$ in the water body beneath lake ice has a median value of −57.9‰ (ranging from −89.6‰ – −26.7‰) (Supplementary Fig. 4a), similar to that of diffusion during ice-free periods on the Tibetan Plateau. The median $\alpha_C$ value of water storage during ice-covered periods is 1.038 ($n = 140$) (Supplementary Fig. 4b), which indicates that acetate fermentation is the primary pathway for $CH_4$ production during ice-covered periods, accompanied by significant oxidation[27,29–31]. This finding further implies that $CH_4$ production continues throughout the winter in the Tibetan Plateau thermokarst lakes. Meanwhile, this pathway aligns with that of $CH_4$ emissions via diffusion during ice-free periods[18], but contrasts with that via ebullition during ice-free periods, which is predominantly driven by $CO_2$ reduction on the Tibetan Plateau[19]. Similarly, high-emission point sources and hotspots in Siberian lakes are primarily driven by $CO_2$ reduction, while lower-emission processes are influenced by acetate fermentation[27]. However, in Western Greenland, it was shown that the primary production pathway for $CH_4$ released through ebullition is acetate fermentation[32], whereas in Finland, $CH_4$ emissions via diffusion are mainly driven by $CO_2$ reduction[33]. The discrepancy is attributed to that the different composition of microbial communities in aquatic systems can control the $CH_4$ production pathways[34,35]. Additionally, the environmental factors, such as temperature[36], salinity[19], and substrates available for methanogenesis[37], can further influence the pathway of $CH_4$ production.

Under future climate scenarios, shorter ice-covered periods and warmer lake water are expected to enhance the proportion of $CH_4$ ebullitive emissions from thermokarst lakes on the Tibetan Plateau. Particularly, changes in lake water temperature and dissolved oxygen content can shift the methanogenic pathway, with more $CH_4$ likely released through ebullition driven by $CO_2$ reduction[36], potentially

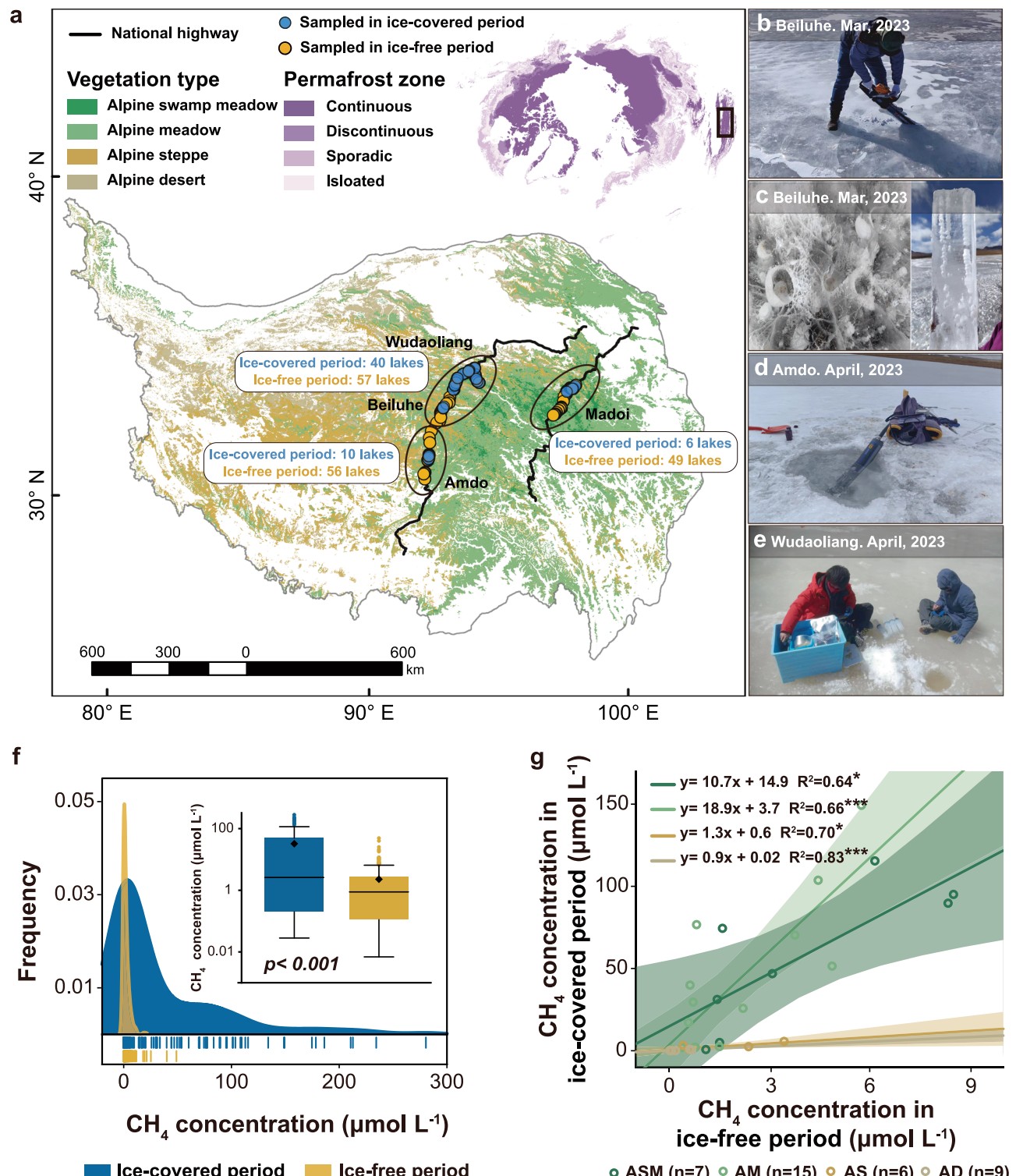

**Fig. 1 | Field observations of thermokarst lakes during ice-covered period on the Tibetan Plateau. a** Distribution of the 56 monitored thermokarst lakes during ice-covered period (this study) and 162 thermokarst lakes during ice-free period[18]. A total of 409 field observations were conducted from June 2019 to April 2023. Permafrost and vegetation distribution data are sourced from existing distribution dataset[12,69]. **b**–**e** The images depict field observations during the ice-covered periods from March to April, captured by P. L and M. M. **f** Density frequency of dissolved $CH_4$ concentrations and their comparison between the ice-covered period and ice-free period. The lines indicate $CH_4$ concentrations of individual samples during ice-free and ice-covered periods. **g** The relationships of dissolved $CH_4$ concentrations between the ice-covered and ice-free periods under vegetation types of alpine swamp meadow (ASM), alpine meadow (AM), alpine steppe (AS) and alpine desert (AD). Notably, the circles represent the individual lakes observed during both the ice-free and ice-covered periods. The corresponding values indicate the average $CH_4$ concentration within the same lake during the two periods. The shaded area represents the 95% confidence interval. *$P < 0.05$, **$P < 0.01$, ***$P < 0.001$.

increasing future total $CH_4$ emissions from thermokarst lakes. Although regional divergence exists in the $CH_4$ accumulation during ice-covered periods[38–40], potentially affecting future $CH_4$ emission prediction, the results shed light on the mechanistic understanding of $CH_4$ release dynamics in the Tibetan Plateau thermokarst lakes. Our study highlights the need to account for winter $CH_4$ production in thermokarst lakes, which may substantially contribute to future annual $CH_4$ emissions under ongoing climate warming.

## CH₄ flux in ice-melting and ice-free periods

To systematically evaluate the total annual $CH_4$ release from thermokarst lakes on the Tibetan Plateau, we quantified the water storage and ice bubble storage by conducting the field observations during ice-cover periods and synthesizing our published fluxes during ice-free period[18] (see Methods). During ice-melting periods, we calculated the proportions of $CH_4$ flux between ice bubble and water storage with different vegetation types (Supplementary Fig. 5). The result shows that during the ice-melting period, $CH_4$ release from thermokarst lakes primarily originates from water storage (Fig. 2a). The median $CH_4$ release flux from water storage is 0.8 (0.01–39.9) g m$^{-2}$ yr$^{-1}$, while $CH_4$ release from ice bubble occurs at a rate of 0.3 (0–17.0) g m$^{-2}$ yr$^{-1}$ (Fig. 2a), corroborating recent observations[38]. Combined, the total $CH_4$ release from ice bubble and water storage ranges from 0.01–56.9 g m$^{-2}$ yr$^{-1}$, with a median of 1.1 g m$^{-2}$ yr$^{-1}$. Similarly, we observed a distribution pattern of $CH_4$ release flux from thermokarst lakes under different vegetation type within the watershed (Fig. 2b). Our study reveals that water storage and ice bubble are estimated to contribute 71.1% and 28.9% to the $CH_4$ release from thermokarst lakes during the ice-melting periods, respectively.

To estimate the proportions of ebullition and diffusion release for annual $CH_4$ emissions, we collected and analyzed the previous data of $CH_4$ emissions during ice-free period[19] (Supplementary Fig. 6). Our results show that the median $CH_4$ release flux is 13.6 (0.1–481.4) g m$^{-2}$ yr$^{-1}$, with 2.0 (0.01–71.5) g m$^{-2}$ yr$^{-1}$ for diffusion and 11.2 (0.1–422.6) g m$^{-2}$ yr$^{-1}$ for ebullition (Fig. 2c). The highest $CH_4$ fluxes are also found in thermokarst lakes under vegetation type of ASM with a median of 30.7 g m$^{-2}$ yr$^{-1}$, followed by AM, AS and AD (Fig. 2d). The proportions of $CH_4$ release from thermokarst lakes through diffusion and ebullition are 15.6% and 84.4%, respectively. By comparing $CH_4$ release flux between the ice-melting and ice-free period, we show that the contribution of $CH_4$ release during ice-melting is essential for the annual estimation, accounting for 5.2–25.8% of the total release from these lakes (Fig. 2e). Although lower water temperatures and reduced external hydrological inputs typically limit the $CH_4$ production, microbial activity continues in both the water column and sediments, even when fully covered by ice[9,41]. Additionally, significant inputs of allochthonous organic matter during the early and late ice-covered periods[42], promote the decomposition of organic matter, leading to a high proportion of $CH_4$ emissions during the ice-melting period. Notably, the lake ice covered period account for approximately 40% for a year on the Tibetan Plateau[43].

## Annual emissions

To estimate the extent of previously underestimated $CH_4$ release on the Tibetan Plateau, we used a Monte Carlo approach to randomly sample thermokarst lake $CH_4$ flux for each vegetation type. We then obtained the total annual $CH_4$ emissions by multiplying the area of thermokarst lakes and mean $CH_4$ flux under each vegetation type. Results indicate that during ice-free period, thermokarst lakes emit $CH_4$ at a rate of 54.2 ± 9.9 Gg C yr$^{-1}$, with 46.9 ± 9.8 Gg C yr$^{-1}$ from ebullition and 7.3 ± 1.4 Gg C yr$^{-1}$ from diffusion. During ice-melting period, $CH_4$ emission is estimated at 11.2 ± 1.6 Gg C yr$^{-1}$, with 3.3 ± 0.6 Gg C yr$^{-1}$ released from bubbles trapped within lake ice, and 7.9 ± 1.5 Gg C yr$^{-1}$ from the water body beneath lake ice. The total annual $CH_4$ emissions from thermokarst lakes on the Tibetan Plateau are estimated to be 65.5 ± 10.0 Gg C yr$^{-1}$. Our study reveals that previously overlooked $CH_4$ emissions during ice-melting period account for 17.1% (13.5–20.7%) of the annual $CH_4$ emissions, with 5.0% (3.8–6.2%) attributed to ice bubbles and 12.1% (9.2–15.0%) from water storage (Table 1, Supplementary Fig. 7a). The contributions of $CH_4$ emissions during ice-melting period to the whole year vary with vegetation types of alpine swamp meadow (25.8%), alpine meadow (16.9%), alpine steppe (11.5%) and alpine desert (5.2%) (Fig. 2e, Supplementary Fig. 7b). Compared with the Arctic region, where the contribution of $CH_4$ release during ice-melting was 20–74%[7,44–46], the proportion of $CH_4$ release during ice-melting period on the Tibetan Plateau is relatively lower. This is attributable to that the Arctic permafrost regions have higher SOC contents[47,48] and longer ice-covered durations[43,45]. Our study shows the magnitude of $CH_4$ emissions during thermokarst lake ice melting on the Tibetan Plateau, providing essential monitoring data for alpine permafrost carbon cycling.

To better illustrate the magnitude of carbon release from thermokarst lakes, we converted the $CH_4$ emissions into $CO_2$ equivalent using the 100-year global warming potential[49]. Result show that $CH_4$ emissions from Tibetan Plateau thermokarst lakes are estimated at 2362.0 ± 360.8 Gg $CO_2$-eq yr$^{-1}$ (Table 1). Remarkably, although the area of thermokarst lakes accounts for only 0.2% of the Plateau permafrost regions[11,15], our estimated total amount of $CH_4$ release from thermokarst lakes offsets approximately 6.4 ± 1.0% of the alpine grasslands carbon sink on the Plateau[50]. Hence, we underscore the $CH_4$ release from thermokarst lakes for the future climate feedback in alpine permafrost region. Despite overcoming challenges such as harsh climatic conditions during the ice-covered period and limited accessibility in high-altitude permafrost regions, our estimation has possible uncertainties due to field observations. On the one hand, the sampling focuses on a specific depth of the water body beneath lake ice, resulting in the vertical difference in $CH_4$ concentrations is neglected. On the other hand, the duration of ice-covered and ice-melting periods was estimated using remote sensing[43], which might not represent all small thermokarst lakes. Therefore, enhancing more observations, especially for changes in lake ice phenology on the Tibetan Plateau, is crucial for a comprehensive understanding of $CH_4$ emissions from thermokarst lakes.

## Expansion of thermokarst lake

To predict future changes of thermokarst lakes on the Tibetan Plateau, we used the Random Forest (RF) model to assess the susceptibility distribution of thermokarst lakes and calculate the actual area of thermokarst lake based on the lake area density under different vegetation types (Supplementary Table 5) (see Methods). The results show that thermokarst lakes on the Tibetan Plateau are expected to increase under SSP scenarios. By 2100, the total area of thermokarst lakes is projected to reach 3912 km$^2$ (increase by 85.9% compared with 2020) under SSP1-2.6, 3926 km$^2$ (86.5%) under SSP2-4.5, and 4102 km$^2$ (94.9%) under SSP5-8.5 (Supplementary Fig. 8). These changes correspond with the previous remote sensing observations that showing an increase of 14.7% per-decade between 1969 and 2010 and 20.0% per-decade between 2010 and 2019[51]. Our results forecast thermokarst lakes area will increase by 33.2% per decade between 2020–2040, which is higher than the previous increasing rate due to future warming and wetting in the alpine region (Supplementary Fig. 9). Although future prediction of thermokarst lakes relies more on the process model and need more verification, abrupt permafrost thaw cannot be simulated by Earth System Models (ESMs). Our results show the response of these lakes on the Tibetan Plateau to future climate change through considering the influencing factors of climate, topography, hydrology, soil, permafrost, and human activity (Supplementary Table 3). Our study highlights the future changes of alpine thermokarst lakes and their ecological and environmental impacts.

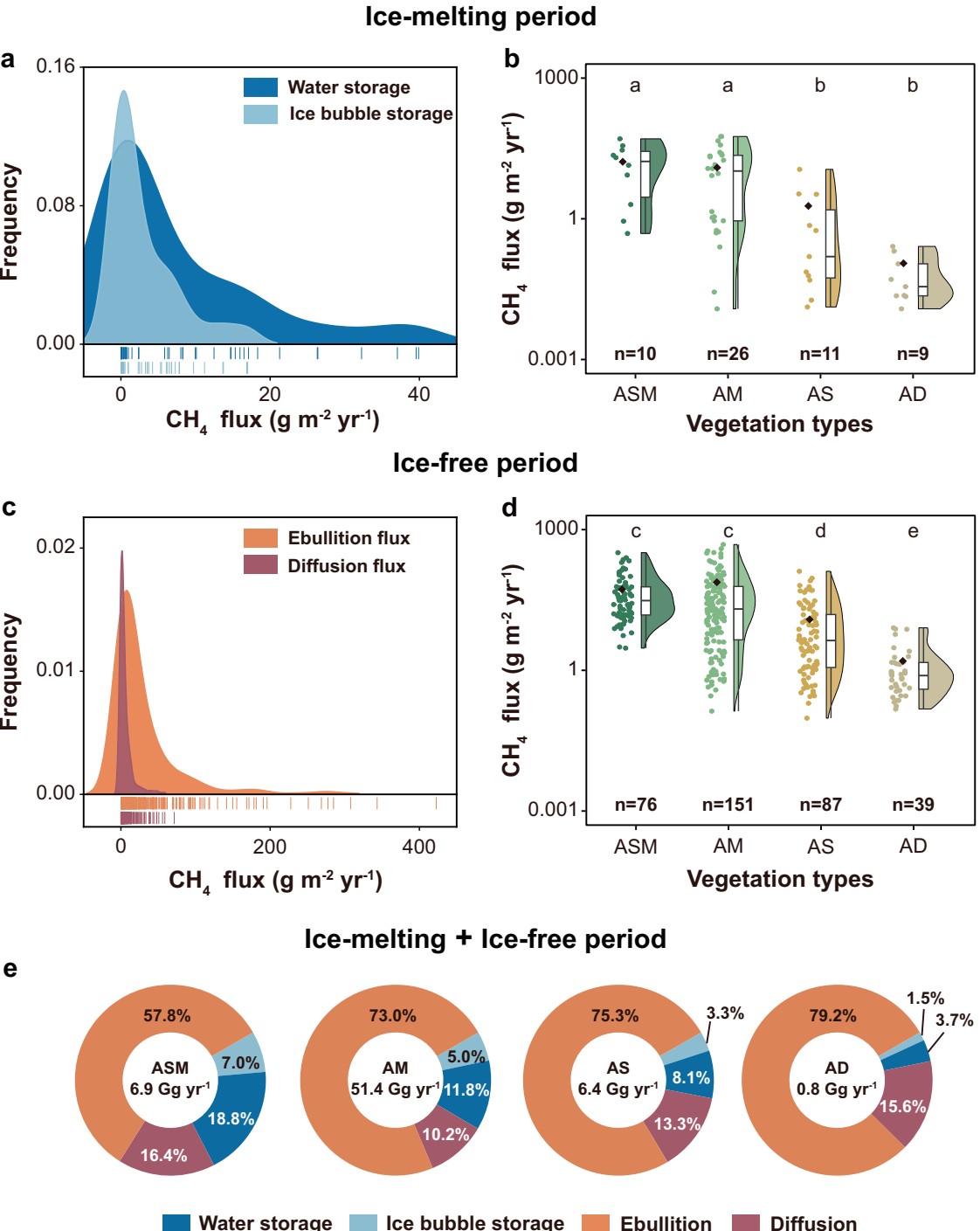

**Fig. 2 | CH₄ emission flux from thermokarst lakes during ice-melting and ice-free periods on the Tibetan Plateau. a** Density distribution of CH₄ release flux from water storage and ice bubble storage during ice-melting period. The lines indicate CH₄ fluxes of individual sampling points from water storage and ice bubble storage. **b** Total CH₄ flux from thermokarst lakes during ice-melting under the vegetation types of alpine swamp meadow (ASM), alpine meadow (AM), alpine steppe (AS) and alpine desert (AD). The violin plot illustrates the magnitude and distribution density of CH₄ fluxes of individual samples, in which the boxes represent the 25th and 75th percentiles; the black line indicates the median value, and the black diamond represents the mean value. **c** Density distribution of CH₄ release flux in diffusion and ebullition ways during ice-free period. The lines indicate CH₄ fluxes of individual sampling points from ebullition and diffusion. **d** Total CH₄ flux from thermokarst lakes during ice-free period with vegetation types. **e** The proportions of annual CH₄ release in the ways of water storage, ice bubble, ebullition and diffusion for the whole year under different vegetation types on the Plateau. The number at centers of the circle indicates the annual CH₄ emissions from thermokarst lakes (Gg C yr⁻¹). One-way analysis of variance (ANOVA) with Tukey's HSD post hoc comparisons are used to test differences in concentrations across different vegetation types and periods at a significance level of $P < 0.05$.

## Future CH₄ emissions

To show the effects of future lake expansion and ice loss on CH₄ release, we projected future CH₄ emissions both with and without considering changes in thermokarst lake ice (see Methods). Our findings indicate significant increases in CH₄ emissions from Tibetan Plateau thermokarst lakes even if lake ice loss is not considered (Supplementary Table 6). Specifically, CH₄ emissions during the ice-melting period are expected to rise by 76.7%–77.8% by 2050 and

**Table 1 | Annual CH₄ and CO₂-equivalent emissions from thermokarst lakes during ice-free and ice-melting periods on the Tibetan Plateau**

| Vegetation types | CH₄ emissions (Gg C yr⁻¹) | | | | Annual flux (Gg C yr⁻¹) | CH₄ emission proportions |
|---|---|---|---|---|---|---|
| | Ice-free period | | Ice-melting period | | | |
| | Diffusion flux | Ebullition flux | Water storage | Ice bubble storage | | |
| ASM | 1.13 ± 0.15 | 3.96 ± 0.53 | 1.29 ± 0.37 | 0.48 ± 0.14 | 6.85 ± 0.68 | 10.49% |
| AM | 5.22 ± 1.35 | 37.52 ± 9.72 | 6.08 ± 1.39 | 2.59 ± 0.59 | 51.41 ± 9.93 | 78.57% |
| AS | 0.85 ± 0.15 | 4.80 ± 0.84 | 0.52 ± 0.29 | 0.21 ± 0.12 | 6.38 ± 0.91 | 9.71% |
| AD | 0.13 ± 0.03 | 0.64 ± 0.14 | 0.03 ± 0.01 | 0.01 ± 0.01 | 0.81 ± 0.15 | 1.23% |
| Total CH₄ emission (Gg C yr⁻¹) | 65.45 ± 10.00 | | | | | |
| CO₂-equivalent emissions (Gg CO₂-eq yr⁻¹) | 2362.02 ± 360.79 | | | | | |

The data was shown as mean ± standard error (SE). Vegetation types include alpine swamp meadow (ASM), alpine meadow (AM), alpine steppe (AS) and alpine desert (AD)

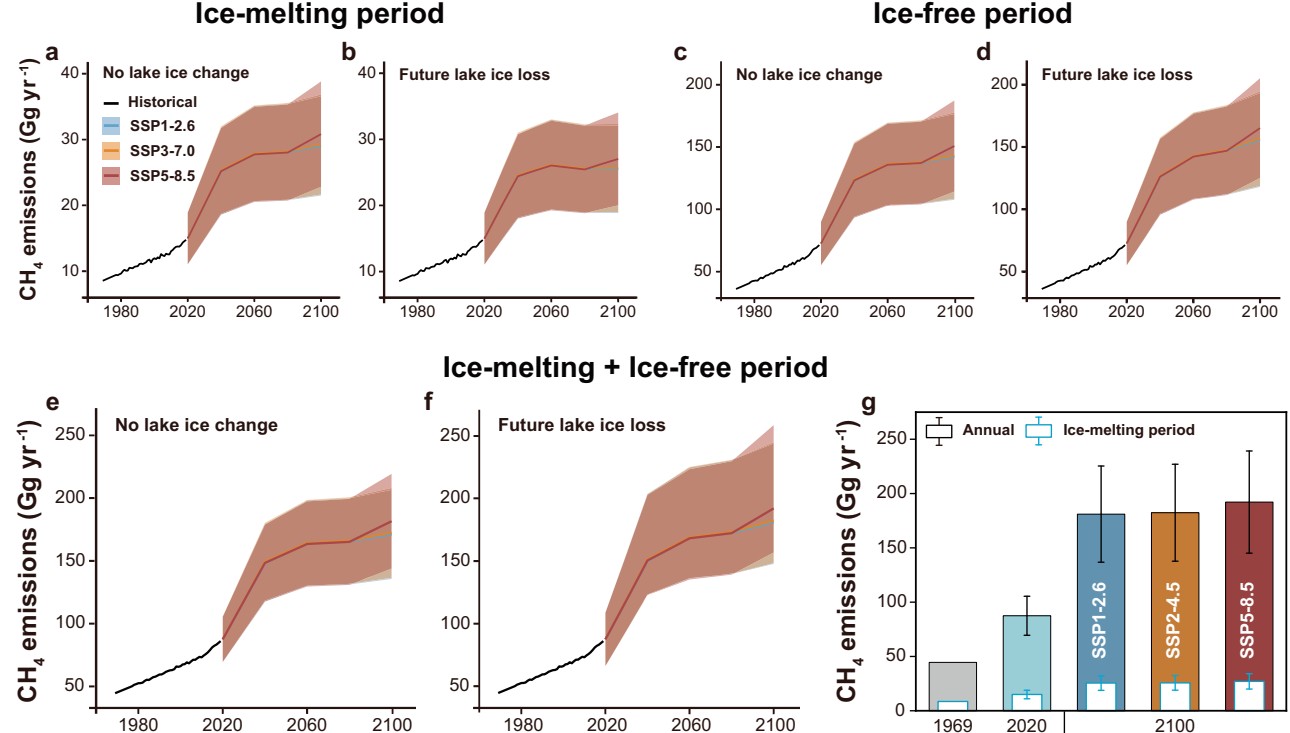

**Fig. 3 | Simulated future CH₄ emissions from Tibetan Plateau thermokarst lakes by 2100.** The projected CH₄ emissions from thermokarst lakes up to 2100 during ice-melting (**a-b**) and ice-free periods (**c-d**) under Shared Socioeconomic Pathways (SSPs). **a**, **c** depict the trends in CH₄ emission from thermokarst lakes without considering the changes of ice-covered duration. Conversely, **b**, **d** show future changes of CH₄ emissions after considering the changes of ice duration, reflecting a dynamic modeling of ice impacting on CH₄ emissions. **e** The predicted annual CH₄ emissions from thermokarst lakes up to 2100, without accounting for changes in the duration of ice cover. **f** The predicted annual CH₄ emissions from thermokarst lakes with future lake ice loss. **g** The bar charts show the past, present, and future CH₄ emissions. The charts with white bars represent CH₄ emissions during the ice-melting period, and color bars show the annual emissions. Data are reported as mean ± standard error (SE).

93.8%–105.8% by 2100 under the SSP1-2.6, SSP2-4.5, and SSP5-8.5 (Fig. 3a). During the ice-free period, the emissions could increase by 95.9%–107.9% by 2100 (Fig. 3c), with total annual emissions potentially reaching 116.7–117.3 Gg C yr⁻¹ by 2050 and 128.0–135.9 Gg C yr⁻¹ by 2100 (Fig. 3e). However, over the past 40 years, the average ice-covered duration on the Tibetan Plateau has decreased at a rate of 0.2 days per year (Supplementary Fig. 10)[43]. Taking the loss of lake ice into account, by 2100, a shorter ice-covered duration could reduce the increase in CH₄ emissions during ice-melting periods but enhance it during ice-free periods (Fig. 3b, d). This is because the shortened ice-covered period means less CH₄ accumulation in the water beneath the

ice[40,52] and more emissions during ice-free periods[53]. Moreover, a longer ice-free period can boost methanogenic activity, further enhancing CH₄ emissions[54,55]. Taken together, future CH₄ emissions from the Tibetan Plateau thermokarst lakes could reach 135.5–143.8 Gg C yr⁻¹ by 2100 (Fig. 3f), and considering the CO₂-equivalent emissions of CH₄, this could potentially offset about 14.1% of the carbon sink in alpine grasslands[50].

The future increase of CH₄ emissions on the Tibetan Plateau is possibly different with the Arctic lakes[7]. This is attributable to the different trends of thermokarst lake areas between the Arctic and Tibetan Plateau, specifically the Arctic is undergoing the extensive

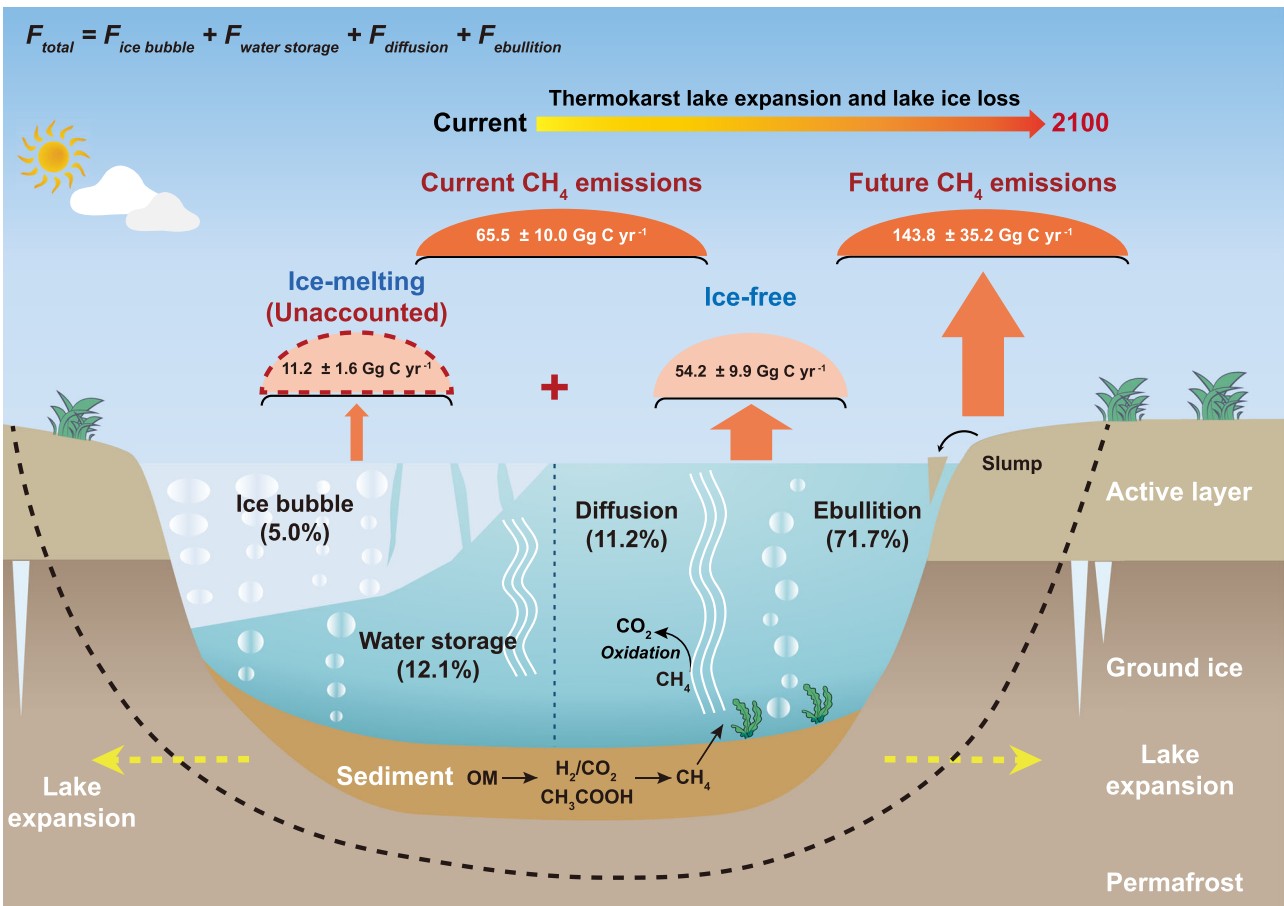

**Fig. 4 | Schematic of annual CH₄ release from thermokarst lakes during ice-melting and ice-free periods on the Tibetan Plateau.** The total annual CH₄ release includes the four parts shown in the formula. During ice-melting, CH₄ release from thermokarst lakes is through both ice bubble and water storage beneath lake ice. During ice-free period, CH₄ release is from diffusion and ebullition. The percent data is from the integrated analysis of field observations, showing the proportion of CH₄ release from the ice bubble (5.0%), water storage (12.1%), ebullition (71.7%) and diffusion (11.2%) for the whole year. The dotted black line represents the future expansion of thermokarst lakes.

draining of thermokarst lakes[56,57], whereas thermokarst lakes on the Tibetan Plateau are expanding rapidly in response to climate change[16,51,58,59]. To date, the estimate of CH₄ release rarely involves the changes in areas and ice phenology of thermokarst lakes. The advantage of this study is that we fully consider the changes of thermokarst lake based on the remote sensing monitoring and future climate scenarios. However, our simulations did not fully incorporate dynamic vegetation changes. The variations of vegetation and permafrost stability highlight that not all thermokarst lakes respond uniformly, underscoring the need for more detailed vegetation change data to improve the projections, particularly under warming-driven shifts in vegetation types. Future increase in CH₄ emissions from thermokarst lakes on the Tibetan Plateau highlights the growing importance in global greenhouse gas inventory data. Our study demonstrates that the expansion of thermokarst lakes and shortening of lake ice-covered duration accelerate CH₄ emissions, playing a crucial role in predicting permafrost carbon feedback to climate change.

Our study offers advantages in simulating CH₄ emissions from thermokarst lakes. Firstly, by integrating the distribution of thermokarst lakes and environmental factors, we predicted the future changes in thermokarst lake area on the Tibetan Plateau. Secondly, although our simulations do not include dynamic vegetation changes expected in Tibetan Plateau permafrost regions with warming, we consider the effects of vegetation and climate on the current distribution of thermokarst lakes in order to reduce the uncertainty in estimating CH₄ emissions. Furthermore, given that thermokarst lakes affect small areas within the permafrost region, changes in vegetation may not be the primary determinant of CH₄ emissions compared to widespread gradual permafrost thaw emissions[3]. Finally, by integrating field observations of CH₄ release, analysis of dissolved δ¹³C-CH₄ and CO₂, and future predictions, we provide current and future changes in previously underestimated CH₄ release during ice-melting from thermokarst lakes (Fig. 4). Our study also identifies additional important knowledge gaps, such as the verification of lake expansion and spatial heterogeneity of lake ice phenology, that will require further research before carbon release from thermokarst lakes can be simulated by ESMs.

### Implications
The exploration into the dynamics of thermokarst lakes and their impact on climate change yields invaluable insights with far-reaching implications for both scientific understanding and proactive mitigation strategies. This study delves into the significance of these research findings and outlines potential strategies to address the challenges posed by thermokarst lake dynamics in the context of climate change. First and foremost, the research on thermokarst lakes underscores their pivotal role in the carbon cycle and climate feedback loop. By quantifying previously underestimated CH₄ emissions, particularly during ice-melting periods, studies have illuminated a critical aspect of permafrost thaw dynamics. This enhanced understanding not only refines our knowledge of regional carbon dynamics but also highlights the urgency of addressing these emissions to mitigate climate change.

One of the key values of this research lies in its contribution to improving ESMs and climate projections. By integrating field observations, machine learning modeling, and projections from climate models, researchers have laid the groundwork for more accurate simulations of thermokarst lake dynamics and their implications for future climate scenarios. This advancement is instrumental in informing climate policy decisions and developing adaptive strategies to mitigate the impacts of permafrost thaw. Furthermore, the insights gleaned from thermokarst lake research offer valuable guidance for policymakers and stakeholders seeking to address the challenges posed by climate change. From enhancing monitoring efforts to implementing targeted mitigation measures, there are several avenues for action. Investing in enhanced monitoring permafrost thaw and remote sensing techniques is essential to capture the spatial variability of thermokarst lakes and track their evolution over time[60,61]. This comprehensive data collection is crucial for refining climate models and improving the accuracy of future projections.

Moreover, proactive mitigation strategies are imperative to mitigate the impacts of thermokarst lake dynamics on the climate system[62]. This includes measures to reduce anthropogenic greenhouse gas emissions[63], as well as targeted interventions to mitigate $CH_4$ emissions from thermokarst lakes[64]. Strategies such as lake management techniques, habitat restoration, and land-use planning can help minimize the release of $CH_4$ and mitigate the climate feedback effects of thermokarst lakes. Research on thermokarst lakes offers invaluable insights into the complex dynamics of permafrost thaw and its implications for global climate change. By enhancing our understanding of $CH_4$ emissions from thermokarst lakes and refining climate models, this research provides a foundation for informed decision-making and proactive climate mitigation strategies. Moving forward, continued investment in research, monitoring, and mitigation efforts is essential to address the challenges posed by thermokarst lakes and safeguard the stability of the global climate system.

## Methods

### Study area and field observations

Permafrost is widely distributed across the Tibetan Plateau, covering approximately 50% of the Plateau[12]. The annual mean ground temperature has increased by 0.43 °C per decade[65], resulting in a significant deepening of the active layer[66–68]. Thermokarst lakes are widely distributed on the permafrost regions and are dominated by lakes with small surface areas (<10,000 m²), which account for ~80% of the total. In this study, we measured the dissolved $CH_4$ concentrations in 56 alpine thermokarst lakes during ice-covered period in 2023. Furthermore, we integrated $CH_4$ concentrations data from 162 thermokarst lakes during ice-free period on the Tibetan Plateau (Fig. 1a)[18], all located within the same study area. The observed lakes are primarily located in the continuous permafrost regions with elevations between 4210 and 5127 m (Fig. 1a). The area of thermokarst lakes ranges from 373 m² to 648,966 m², with the majority being less than 100,000 m² (Supplementary Fig. 1b, Supplementary Table. 1).

Based on the dataset of vegetation type distribution on the Tibetan Plateau[69], the observed thermokarst lakes are mainly distributed in the regions with four vegetation types: alpine swamp meadow (ASM), alpine meadow (AM), alpine steppe (AS) and alpine desert (AD). These regions exhibit distinct differences in vegetation composition, soil characteristics, and organic carbon content. ASM is characterized by sedges and rushes, with saturated soils rich in organic matter and high moisture levels, which supports significant carbon sequestration[70,71]. AM features grasses and herbs, with well-drained loamy soils and moderate organic carbon content, experiencing seasonal freeze-thaw cycles[71]. AS comprises drought-tolerant grasses and shrubs, with dry, coarse soils and low organic carbon content, showing significant spatial variability[71]. AD is characterized by sparse vegetation, very dry rocky soils, and minimal organic carbon, with high pH

levels and significant wind erosion[72]. During ice-covered period from March to April 2023, the dissolved $CH_4$ in lake water was collected using the headspace equilibrium method (Fig. 1e)[73,74]. In each lake, 3–6 sampling points were distributed from the edge to the center of lake. Briefly, 400 mL of surface water from a depth of 15 cm was collected with a 500 mL syringe after removing surface ice with a chainsaw (Fig. 1b), Subsequently, 100 mL of pure $N_2$ was injected into the syringe, and the syringe was immediately shaken for 3 minutes to equilibrate the headspace in the field. A 100 mL of the headspace gas was transferred from the syringe into a specialized aluminum foil gas sampling bag, which was then sealed for preservation. At each sampling point, two gas samples were collected for parallel comparison. The sample collection techniques employed during ice-free period[19], were in line with the methodologies used in this study. The collection period for $CH_4$ during ice-free period spanned from May to October in the years 2019 –2021. The concentration of dissolved $CH_4$ was measured using a gas chromatograph, with a measurement precision of ±2.2%.

At each sampling point, we also measured water quality parameters, including water temperature, dissolved oxygen and conductivity, using a multi-parameter water quality meter (Fig. 1d). Meteorological data, including air temperature, atmospheric pressure, and wind speed, were sourced from automated field weather stations located at Liangdao River, Tanggula, and Beilu River, as well as from the National Meteorological Science Data Center (NMSDC; http://data.cma.cn/) for the Maduo, Wudao Liang, and Tuotuo River.

### Calculation of dissolved $CH_4$ concentrations

Dissolved $CH_4$ concentration in water was calculated by integrating the principles of the ideal gas law and Henry's law with the concentration of $CH_4$ in the headspace of the syringe[18,19,75,76]. This methodological approach allowed for the precise calculation of $CH_4$ levels in the aquatic environment, providing accurate results for gas solubility in lake water under varying atmospheric and temperature conditions. The dissolved $CH_4$ concentration in water was calculated using Eq. (1):

$$C_w = \frac{(C_h \times V_h + S \times V_w)}{V_w^{-1}} \tag{1}$$

where $C_w$ is dissolved $CH_4$ concentration in water (μmol L⁻¹). $S$ is the $CH_4$ solubility in water when equilibrium is reached between the air and water phases (μmol L⁻¹)[77]. $V_h$ is the volume of headspace gas in the syringe (L). $V_w$ is the volume of water in the syringe (L). $C_h$ is $CH_4$ concentration in the headspace gas of the syringe, calculated using Eqs. (2) and (3).

$$C_h = \frac{n}{V_h} \tag{2}$$

$$n = \frac{P_m \times V_h}{R \times T_{air}} \tag{3}$$

where $n$ represents the molar quantity of $CH_4$ in the headspace gas when equilibrium is reached between the air and water phases (mol). $P_m$ is $CH_4$ partial pressure in the headspace gas (atm), determined by the $CH_4$ volume concentration in the headspace gas and the actual atmospheric pressure at the sampling location. $R$ is the ideal gas constant (L atm K⁻¹ mol⁻¹). $T_{air}$ is the air temperature in degrees Kelvin (K)[22].

### Calculation of $CH_4$ flux during ice-free period

Based on the collected $CH_4$ concentration data during the ice-free period, a thin boundary layer method was used to calculate the $CH_4$ diffusion fluxes from thermokarst lakes[18,19,22]. It is important to note that the data measured in this study is the same with our previous study.

Consequently, this study did not account for potential differences that might arise from variations in sampling or calculation methods between data from ice-free period and ice-covered period. The diffusion fluxes from surface water to the atmosphere are correlated with the concentration gradient of $CH_4$ between the water and the atmosphere, as well as the gas transfer velocity[78]. The diffusion fluxes were calculated using Eq. (4):

$$F_{\text{diffusion}} = K \times (C_{w_{\text{ice-free}}} - C_{eq_{\text{ice-free}}}) \times n_{\text{ice-free}} \times m \qquad (4)$$

where $F_{diffusion}$ is the $CH_4$ diffusion fluxes from surface water to the atmosphere during ice-free period (g m$^{-2}$ yr$^{-1}$). $K$ is the gas transfer velocity (cm h$^{-1}$). $C_{w_{\text{ice-free}}}$ is the concentration of dissolved $CH_4$ in water during ice-free period (μmol L$^{-1}$)[79]. $C_{eq_{\text{ice-free}}}$ is the dissolved $CH_4$ concentration in water at equilibrium with the atmosphere (μmol L$^{-1}$)[77]. The $CH_4$ concentration in the atmosphere was obtained from the Waliguan Baseline Observatory (https://gaw.kishou.go.jp/). $n_{\text{ice-free}}$ is the duration of the ice-free period (200 days)[43]. m is the molar mass of $CH_4$, which is 16.04 g/mol.

The gas transfer velocity is correlated with wind speed and water temperature, it was calculated using Eqs. (5) and (6):

$$K = \left(\frac{S_c}{600}\right)^{-X} \times K_{600} \qquad (5)$$

$$S_c = 1897.8 - 114.28 T_{\text{water}} + 3,2902 T_{\text{water}}^2 - 0.039061 T_{\text{water}}^3 \qquad (6)$$

where $S_c$ is the Schmidt number of $CH_4$[80]. $T_{water}$ is water temperature (°C). If the wind speed is at most 3 m/s, $x = 0.66$; for wind speeds exceeding 3 m/s, $x = 0.5$[81]. $K_{600}$ refers to the value of $K$ when the Schmidt number is 600.

Different studies employ various methods to calculate this value, leading to discrepancies. To reduce uncertainty, this study adopts the average value derived from three different calculation methods[82–84]. The calculation is given in Eqs. (7) and (8).

$$K_{600} = \frac{(2.07 + 0.215 U_{10}^{1.7}) + 0.45 U_{10}^{1.64} + (1.68 + 0.228 U_{10}^{2.2})}{3} \qquad (7)$$

$$U_{10} = U_Z \times \left(1 + \frac{C_{d10}^{0.5}}{K} \times \ln\frac{10}{Z}\right) \qquad (8)$$

where $U_{10}$ represents the wind speed at a height of 10 m (m/s). $U_z$ represents the wind speed at a height of Z m (m/s). In this study, the wind speed data, measured at a height of 2 m, was obtained from automated field weather stations and the National Meteorological Science Data Center. $C_{d10}$ is the drag coefficient measured at 10 meters above the ground, and $K$ denotes the von Kármán constant.

Through Eqs. 1 to 8, the fluxes of $CH_4$ emission from surface waters into the atmosphere during ice-free period were determined. Besides diffusion, ebullition is another primary pathway for $CH_4$ emissions from thermokarst lakes[22,46]. Compared to diffusion, ebullition is typically intermittent and can exhibit significant fluctuations in $CH_4$ emissions over short periods. However, it contributes substantially to the total $CH_4$ emissions from these lakes[19,22,85]. We collected published data on the proportion of $CH_4$ emissions via ebullition from thermokarst lakes on the Tibetan Plateau. Additionally, the proportion of ebullition in $CH_4$ emissions does not show significant differences across regions with different vegetation types ($P > 0.05$)[19]. We calculated the average proportion of $CH_4$ emissions via ebullition from thermokarst lakes under the vegetation type of alpine swamp meadows ($n = 48$), alpine meadows ($n = 52$), and alpine steppe ($n = 20$). Since there was no data available for alpine desert, the average value

from the other three vegetation types was used (Supplementary Fig. 6). The ebullition flux during ice-free period from thermokarst lakes under different vegetation types was calculated using Eqs. (9) and (10):

$$F_{\text{ebullition}} = \frac{M \times F_{\text{diffusion}}}{1 - M} \qquad (9)$$

$$F_{\text{ice-free}} = F_{\text{diffusion}} + F_{\text{ebullition}} \qquad (10)$$

where $M$ represents the proportion of total $CH_4$ emissions through ebullition corresponding to the type of vegetation underlying the lake. $F_{bubblition}$ is the $CH_4$ ebullition flux during ice-free period (g m$^{-2}$ yr$^{-1}$), $F_{ice-free}$ is the total $CH_4$ flux from thermokarst lakes during ice-free period (g m$^{-2}$ yr$^{-1}$).

### Calculation of $CH_4$ flux during ice-melting period

The ice cover on thermokarst lakes acts as a barrier during frozen periods, preventing $CH_4$ release into the atmosphere. As the lake ice melts during the spring season, the accumulated $CH_4$ is released at a higher flux[86]. The sealed $CH_4$ can be divided into two segments during ice-covered period: water storage and ice bubble storage[7,10,81,87,88]. Water storage refers to dissolved $CH_4$ in water, remaining confined within the lake water until the ice breaks. During this process, a portion of the $CH_4$ undergoes oxidization, with roughly half of it being transformed[87]. During ice-melting period, $CH_4$ is released into the atmosphere through diffusion. Throughout these periods, as the $CH_4$ concentration in the lake water steadily decreases, the diffusion flux correspondingly diminishes as well. When calculating the $CH_4$ flux, we consider the 14 days before and after ice-free period as the high and low emission periods[76]. $CH_4$ flux from water storage were calculated using Eqs. (11), (12), and (13):

$$F_{\text{water storage}} = (F_{\text{high emission}} + F_{\text{low emission}}) \times 14 \qquad (11)$$

$$F_{\text{high emission}} = K_{\text{high}} \times (C_{w_{\text{ice-covered}}} - C_{eq_{\text{ice-covered}}}) \times m \qquad (12)$$

$$F_{\text{low emission}} = K_{\text{low}} \times (C_{w_{\text{ice-melting}}} - C_{eq_{\text{ice-melting}}}) \times m \qquad (13)$$

where $F_{water\ storage}$ represents the $CH_4$ flux from water storage (g m$^{-2}$ yr$^{-1}$). $F_{high\ emission}$ is the $CH_4$ flux during high emission periods (g m$^{-2}$ yr$^{-1}$). $F_{low\ emission}$ is the $CH_4$ flux during low emission periods (g m$^{-2}$ yr$^{-1}$). $C_{w_{\text{ice-covered}}}$ and $C_{w_{\text{ice-melting}}}$ are dissolved $CH_4$ concentration in water at the end of ice-covered period (late April) and the end of ice-melting periods (early May) (μmol L$^{-1}$), respectively. $C_{eq_{\text{ice-covered}}}$ and $C_{eq_{\text{ice-melting}}}$ are dissolved $CH_4$ concentration in water when equilibrium with the atmosphere (μmol L$^{-1}$)[77]. For lakes that were sampled during ice-covered period but lacked data for the end of ice-melting period, we employed an interpolation method to estimate the missing data. This method involved establishing linear relationships between dissolved $CH_4$ concentrations at the end of ice-covered period and those at the end of ice-melting period across various vegetation types in thermokarst lakes.

The formation process of ice bubble storage is as follows: Bubbles released from the lakebed sediment cannot escape into the atmosphere due to the barrier created by the lake ice. Instead, they are gradually encased by the ice as it grows from the top down. Before being completely sealed within the lake ice, approximately 80% of this $CH_4$ ultimately dissolves back into the lake water. The remaining 20% remains trapped within the lake ice in the form of bubbles[87] and is directly released into the atmosphere during the early stages of ice melting. By integrating the principles of ice bubble storage and water storage, and considering the proportion between ebullition and diffusion in thermokarst lakes, a simple numerical approach was used to

estimate $CH_4$ production and storage during ice-covered period. The derivation process for the numerical relationship between water storage and ice bubble storage is outlined in Eqs. (14), (15), and (16):

$$S_{ebullition} = \frac{M \times S_{diffusion}}{1 - M} \quad (14)$$

$$S_{water\ storage} = 0.5(S_{diffusion} + 0.8 S_{ebullition}) \quad (15)$$

$$S_{ice\ bubble\ storage} = 0.2 S_{ebullition} \quad (16)$$

where $S_{ebullition}$ and $S_{diffusion}$ are the total $CH_4$ production through bubbling and diffusion from the thermokarst lakes during ice-covered period, respectively. $S_{water\ storage}$ is the total $CH_4$ emission from water storage. $S_{ice\ bubble\ storage}$ is the total $CH_4$ emission from ice bubble storage. We obtained the numerical relationship between $S_{water\ storage}$ and $S_{ice\ bubble\ storage}$, as shown in Eqs. (17) and (18). (Supplementary Fig. 5):

$$S_{ice\ bubble\ storage} = \frac{2M}{5 - M} \times S_{water\ storage} \quad (17)$$

$$F_{ice\ bubble\ storage} = \frac{2M}{5 - M} \times F_{water\ storage} \quad (18)$$

where $F_{ice\ bubble\ storage}$ is the $CH_4$ fluxes from ice bubble storage during ice-melting period (g m$^{-2}$ yr$^{-1}$). M represents the proportion of $CH_4$ emissions through ebullition from thermokarst lakes, corresponding to the type of vegetation.

The total $CH_4$ fluxes during ice-melting period (g m$^{-2}$ yr$^{-1}$) were calculated using Eq. (19):

$$F_{ice-melting} = F_{water\ storage} + F_{ice\ bubble\ storage} \quad (19)$$

## Pathway of $CH_4$ production

To evaluate the pathway of $CH_4$ production, we measured the $\delta^{13}C$ isotope abundance of both $CO_2$ and $CH_4$ and calculated the apparent fractionation factor ($\alpha_C$). Stable carbon isotope analysis of $CO_2$ and $CH_4$ was conducted using an isotopic ratio mass spectrometry. The $\alpha_C$ was calculated from $\delta^{13}C$ of $CH_4$ and $CO_2$ using Eq. (20)[18]. The $\delta^{13}C$ isotope abundance of $CO_2$ and $CH_4$ released from diffusion and ebullition during ice-free period were analyzed from published data[18,19].

$$\alpha_C = \frac{\delta^{13}CO_2 + 1000}{\delta^{13}CH_4 + 1000} \quad (20)$$

If $\alpha_C$ is between 1.040 and 1.055, it suggests that $CH_4$ production in thermokarst lakes is mainly through acetate fermentation (AM); and $\alpha_C$ is greater than 1.055, it indicates that $CH_4$ production is primarily driven by $CO_2$ reduction[27,28]. When $\alpha_C$ is less than 1.040, it indicates that $CH_4$ oxidation is the dominant process in thermokarst lakes[29].

## Regional upscaling

Based on 409 $CH_4$ fluxes data during ice-free and ice-melting period, this study uses Monte Carlo analysis to estimate the annual $CH_4$ emissions and their uncertainties from thermokarst lakes on the Tibetan Plateau. For each vegetation type (including alpine swamp meadow, alpine meadow, alpine steppe, and alpine desert)[69], 1000 iterations were performed for both the ice-melting period and ice-free period. In each iteration, a random value was sampled from the thermokarst lake $CH_4$ flux data and multiplied by the total area of thermokarst lakes for each vegetation type[15]. Finally, the annual $CH_4$ emissions from thermokarst lakes under different vegetation type

were calculated to determine total $CH_4$ emissions. We used the mean value of the results from the Monte Carlo analysis as the total $CH_4$ emissions from thermokarst lakes on the Tibetan Plateau.

## Thermokarst lake susceptibility

We analyzed a total of 161,341 thermokarst lakes[15]. Prior to machine learning modeling, we used the ENMTools to filter out redundant samples, ensuring that only one lake was present in each 0.0083° × 0.0083° grid. Ultimately, we selected 71164 thermokarst lakes as training samples for the model. Meanwhile, we selected nine environmental factors influencing the distribution of thermokarst lakes. These factors include topography, hydrology, soil, human activities, permafrost and climate (Supplementary Table 3). The topographic factors, including elevation and slope, was derived from the high-precision Global Digital Elevation Model (DEM) (ETOPO Global Relief Model, National Centers for Environmental Information (NCEI) (noaa.gov)). The elevation data was directly obtained from the DEM, while the slope was calculated based on the DEM. The Topographic Wetness Index (TWI) was calculated using DEM[89]. Soil factors, including sand and silt, were obtained from the Harmonized World Soil Database version 2.0 (https://doi.org/10.4060/cc3823en). The human footprint was from Last of the Wild, v2[90]. To simulate the current susceptibility distribution of thermokarst lakes, climatic factors, including monthly maximum air temperature and precipitation, were obtained from WorldClim. The permafrost factor of active layer thickness (ALT) was sourced from a published dataset[91].

## Prediction of thermokarst lake

To predict future changes in thermokarst lake susceptibility under SSP1-2.6, SSP2-4.5, SSP3-7.0, and SSP5-8.5, we used monthly air temperature, precipitation (https://www.worldclim.org/), and ALT[92] data from five general circulation models (GCMs). We refined the resolution to 1 km using the delta approach. All data were masked to the study area and resampled to a spatial resolution of 0.0083° × 0.0083°. Our study used six machine learning models to assess the susceptibility of thermokarst lakes: Random Forest (RF), Generalized Additive Model (GAM), Generalized Boosted Regression Model (GBM), Classification and Regression Tree Analysis (CTA), Artificial Neural Network (ANN), and MaxEnt. We evaluated the performance of these models using Receiver Operating Characteristic (ROC) curves, Kappa statistics, and true skill statistics (TSS) (Supplementary Table 4). Based on these evaluations, we select the RF model, which performed best, for modeling. We set thresholds at 0.2, 0.4, 0.6, and 0.8 to classify susceptibility into five levels: very low, low, medium, high, and very high[93]. We identified the regions with moderate to very high levels as thermokarst lake affected areas. Then we estimated the lake area density (i.e., the area of lakes within a pixel) to calculate the future actual area of thermokarst lakes under different vegetation types. The standard error of the area density was used to quantify the uncertainty in these future projections (Supplementary Table 5).

## Statistical analyses

In this study, we utilized one-way analysis of variance (ANOVA) with Tukey's HSD post hoc comparisons to test differences in dissolved $CH_4$ concentrations and emission fluxes in thermokarst lakes among different vegetation types and during different periods at a significance level of $P < 0.05$. Regression analysis was employed to establish a linear relationship of dissolved $CH_4$ concentrations between the end of ice-covered and ice-melting period. This approach was used to interpolate concentration for lakes in our dataset lacking on-site sampling at the end of ice-melting period. These statistical analyses were performed using Python 3.11 (available at https://www.python.org/). The regional estimates from Monte Carlo analysis were carried out using R v4.2.3 (available at https://www.R-project.org/).

## Data availability

Source Data are provided with this paper. All data supporting the findings are available in the Figshare data repository (https://doi.org/10.6084/m9.figshare.28236848) and Supplementary Information. Source data are provided with this paper.

## Code availability

The codes are also accessible through the same link documented in data availability.

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

## Acknowledgements

This work was supported by the National Key Research and Development Program of China (2024YFF0810900), the National Natural Science Foundation of China (42371132, 42161160328, 42201136), and the Gansu Science and Technology Program (23JRRA1171, 23ZDFA017, 25JRRA647), the Fundamental Research Funds for the Central Universities (lzujbky–2023-eyt01).

## Author contributions

C.M. designed the study, conducted all analysis, interpreted the results and wrote the paper, and all authors contributed to the discussion and revision of the paper. P.L. conducted all the field surveys and some analyses. M.M., C.Z., Z.Z, J.S., Y.J and C.F. contributed the observation and experiments, X.P., G.Z., and L.W. provide environmental data, D.L., C.S., G.W., Z.Z., and Y.Y contributed to the discussion of the study.

## Competing interests

The authors declare no competing interests.
