## [Peer Review File · Nature Communications]

Methane emissions from thermokarst lakes must emphasize the ice-melting impact

Corresponding Author: Professor Cuicui Mu

Version 0:

Reviewer comments:

Reviewer #1

(Remarks to the Author)

The title of the manuscript summarizes the key results:

- The ice melting period is very important in northern lakes and deserves the attention given to it by the authors.
- Ice melting period has to be included in annual emission estimates accompanied by lake ice bubble and water storage estimates.

Yang et al (2023, Nature Communications) focused on CH₄ emissions from thermokarst lakes on the Tibetan Plateau during ice free period. The first author, Guibiao Yang, is also a co-author of the submitted manuscript. The submitted manuscript demonstrates the importance of methane emission during the ice melting period.

Permafrost, thermokarst lakes and ice melting period represent not the easiest possible environment to carry out experimental work. Warming climate has resulted in shorter ice cover duration and changing freeze-up and break-up times in northern lakes. Changes in lake ice phenology, influencing the duration of ice cover, are undergoing significant alterations resulting in shorter ice cover period.

I could not identify major flaws.

Sampling is challenging during the ice melting period. Consequently, only few papers have reported methane emission estimates based on sampling in late winter/during ice melting, further including both bubbling and diffusion of methane. The results presented are interesting widely to scientific community. Methane is a strong greenhouse gas and updated estimates are needed in order to improve the reliability of emission estimates.

The validity of the approach, quality of the data and quality of presentation including supplementary information are good. Further, the reporting of data and methodology are sufficiently detailed and transparent to enable reproducing the results.

Use of statistics and treatment of uncertainties, the appropriateness of statistical tests, and the accuracy of the description of error bars and probability values are satisfactory.

The conclusions and data interpretation are robust, valid and reliable enough. The reliability of methane emission estimates will be improved when more experimental measurements have been gathered during winter and spring ice melt period.

Extrapolation of results is satisfactory. Previous literature has been mostly appropriately cited, beyond US and Sweden, some Canadian and Finnish studies might be useful. The future scenarios include both predictions for future expansion of thermokarst lake area and ice duration.

Well written manuscript including interesting results – it is editorial decision whether methane is so important compared to other topics. The group has long term experience on this issue and has published also in Nature Communications.

Reviewer #2

(Remarks to the Author)

The manuscript titled “Methane emissions from thermokarst lakes must emphasize the ice-melting impact” reveals an enormous work done to expand our knowledge on remote ecosystems and their often-neglected impact on the global processes. The manuscript is based on 3 types of data – 1. Fieldwork data collected in harsh conditions, in remote areas during ice-out season, 2. Compiled data from previously available studies, and 3. Modelling based on future climatic scenarios. The combination of all three data sets is a great strength of this manuscript, and I would like to congratulate the authors with the all the hard work done to achieve these results. I, however, have some concerns about the presentation of the data, especially since articles published in Nature journals are directed to the wide public and not only one specific group of specialists. While the methodology presented by the authors seems sound and valid, with all the limitations we all face in any field of science, I felt that the presentation of the results and, especially, the discussion of what they tell us could be ameliorated before presenting to the wide auditorium of the journal; hence, I would recommend a major review of the manuscript.

My main concern is that the authors present a wide range of data, but it often feels disconnected from each other. With the way the text is presented, I was sometimes confounded as of the need of certain measurements or analyses – when reading the manuscript, I was periodically led to doubting of the need and meaning of presented results. The messages for the reader could be done more carefully, with clear intent, and a continuous storyline. Below I provide my comments in more detail, but most of them deal with this major issue. I would like to underline that I consider this study of great value, and all the results individually have their importance. The manuscript would significantly increase its value and accessibility if the authors were to carefully reconsider their discussions and try to weave one but complete story.

Furthermore, the manuscript presents a study of regional importance - although Tibetan Plateau is a significant high-altitude permafrost area, and the global scientific community will be interested in learning about such rare findings from this region, it must be clearly acknowledged throughout the manuscript that we are dealing with the results from Tibetan Plateau. The authors did precise this in many instances, but there are whole paragraphs when it is unclear whether we are dealing with interpretations for the thermokarst lakes in general, or only those studied in Tibet.

Lines 132-136. I understand the reason the authors started their interpretations this way; however, I feel it is unnecessary, especially since they do not have much to link the methane concentrations with the vegetation itself. I would recommend skipping the comments on the vegetation type and go directly to the gradient of soil water content (from wet meadows to deserts) and the amount of soil organic matter available around the studied ponds, both of which correspond closely to the presented vegetation type gradient. Wet meadows accumulate more undegraded peat, as wet soils are more likely to form persistent anoxic conditions slowing down the biodegradation, and deserts have less organic matter as there are less vegetation. You discuss most of this already in the next lines of this paragraph, and define the ground and organic matter properties of the studied regions in methods, but this information with different landscapes is worthy longer discussion attaching the soil properties (humidity, organic matter, sensitivity to permafrost thaw). To be sure I am not misunderstood, I do not ask to remove the vegetations types from the study, but to treat them as different landscape with different properties in more wholesome ways; meanwhile discussion based on the plant species themselves is not useful the way it is presented right now.

Fig. 1g. It is not clear from how many observations per vegetation type these regressions are based on – it seems that sometimes there are only two points, or so. If the number of observations is that small, I do not consider it worthy to be placed in one of the main figures. This kind of information may be considered as a curiosity (maybe as a supplementary figure?) but not much can be said from it. If I am mistaken, please provide more information, or add all the data points into the graph to make it more informative.

Paragraph starting with the line 142. Isotopic data may often add important insights into methane-related studies; however, in this case I do not see your data as adding any new value to serve your objectives. The authors may want to consider rewriting this paragraph to make it less descriptive-only and add more interpretations, and comparisons with other studies done in winter and summer time in Finnish boreal lakes, or Canadian Arctic lakes. What do these isotopic values tell us about the present and future methane emissions, methane sources? How does the fact that ebullition from these thermokarst lakes come from CO₂ reduction affect our understanding of future predictions? Do we expect these values to change with time?

Line 164. “...we quantified the fluxes through diffusion and ebullition during ice-free period...” I do not consider it fitting to declare that the authors quantified the fluxes, when all the ebullition data in this study, if I understood well, comes from other previously published works. It is ok to use the other data, but in that case the discussion needs to be appropriately constructed. The strength of this study is the data from the ice-out period, which should be put forward, and only then compared to the open-water data available in previous studies. I consider it sufficient to mention the ice-free period emissions as you do with the beginning of the next paragraph (line 180), it could be your first mention there.

Line 171. “We found that CH₄ release from thermokarst lakes was mainly from water storage during ice-melting”. The structure of the sentence leads to diverging interpretations – do the authors mean that the diffusive emission from water storage is the main pathway of methane evasion, when considering only the ice melt period? Or is it rather considering the whole annual emissions? Please rephrase to be more precise.

Lines 253-266. Any reader appreciates honest presentation of study limitations, and as someone working in high Arctic lakes, I understand very well the constraints. I do not know, however, if this section needs so many excuses – these constraints are normal and expected. It would be enough to briefly mention that harsh and remote fieldwork conditions limit

how spatially and temporally representative your data is. I would avoid calling them “unexpected uncertainties”, though, as anyone could expect uncertainties from a couple of samples for such vast areas the study represents. To add to your list of uncertainties, I would like to see the acknowledgement of limitation from sampling depth perspective – if I understood well, you only took one sample from under ice, neglecting vertical differences in water column.

Paragraph starting with line 274. The first half of this paragraph describes the methodology (some details are fine, I imagine, but now it seems to be the main focus, in the result-discussion section). Then the authors move on to saying that their results are comparable to already existing observations. I do not wish to reduce the importance of modeling and the work the authors did to achieve these results, but in order for this whole section to be relevant, it needs to be presented differently. Maybe instead a couple of sentences within the sections on future emissions, or the perspectives?

Section on future methane emissions (from line 301). While the title of the section is promising, I found it rather dry (lots of statistics and future projections in percentages of increase; information that can be presented in a table instead of two paragraphs) and sometimes confusing. I do not wish to discourage the authors; however, the presented results are interesting and useful, but the discussion of their implications was often missing. I would recommend to discuss aspects like: 1. what the shorter period of ice cover would mean for future methane winter storage and emissions – more lake expansion?, less methane accumulation?, more ebullition during longer summers?; 2. How do future prediction vary based on the vegetation types? Do all lakes behave the same? Is it possible to model lake expansion or methane emission based on regional soil/vegetation type?; 3. How do these predictions for Tibetan Plateau differ from other permafrost regions in the world?

Version 1:

Reviewer comments:

Reviewer #2

(Remarks to the Author)

Once again, I would like to congratulate the authors for their hard work in collecting and processing the data and writing this manuscript. My concerns, expressed in the previous round of reviews, have been answered, and I would generally approve the publication of the manuscript with minor recommendations expressed below:

1. When reading the limited publications that touch winter methane story in ice-covered lakes, it is not unusual to see that methane concentrations decrease through winter, assumingly, due to the combination of slower rates of methanogeny (lower temperatures, less labile organic matter) and continuous work of methanotrophs. In the lakes you sampled, it seems to not be the case; nevertheless, it would further improve your discussion if you addressed this divergence in methane (vs CO₂) accumulation under ice. Do you have any suggestions, why lakes in Tibet continue producing CH₄ throughout winter? Can this be answered comparing isotopic data between your and other winter studies? How would such divergence in winter methane stories affect the future models and their interpretations?

2. Lines 130-141. ..."DOC provides the crucial substrate for microbial CH₄ production in the water column"... from my experience, it is rather unusual (however, not unbelievable) that methane correlates with DOC... as methanogeny mostly happens in the sediments (it has been proven that some of it may occur in water column as well; but for shallow lakes I would still lean towards blaming the sediments) DOC and CH₄ concentrations could correlate but not necessary due to direct effects. Methanogens are often presented as sensitive to organic matter quality. I would recommend checking more studies regarding the topic (I can see now that you cite two studies from the same regions, one of which is based on rivers). In short, while DOC and CH₄ correlation is not surprising, I would recommend using a different approach when explaining it.

3. A general recommendation to re-evaluate the main title of the manuscript. While it somewhat fits your study design, I think you could make it less limited (it sounds as a paper describing recommendations for future methodologies in the field; and your implication might go beyond the ice-melting period, even if it represents "only" 17% of annual emissions), but also less generic (you studied thermokarst lakes on Tibetan Plateau, and as we have already discussed, the results do not represent thermokarst lakes globally).

Response to Reviewers of NCOMMS-24-50735

Reviewer #1 (Remarks to the Author):

The title of the manuscript summarizes the key results:

- The ice melting period is very important in northern lakes and deserves the attention given to it by the authors.
- Ice melting period has to be included in annual emission estimates accompanied by lake ice bubble and water storage estimates.

Yang et al (2023, Nature Communications) focused on CH₄ emissions from thermokarst lakes on the Tibetan Plateau during ice free period. The first author, Guibiao Yang, is also a co-author of the submitted manuscript. The submitted manuscript demonstrates the importance of methane emission during the ice melting period.

Permafrost, thermokarst lakes and ice melting period represent not the easiest possible environment to carry out experimental work. Warming climate has resulted in shorter ice cover duration and changing freeze-up and break-up times in northern lakes. Changes in lake ice phenology, influencing the duration of ice cover, are undergoing significant alterations resulting in shorter ice cover period.

I could not identify major flaws.

Sampling is challenging during the ice melting period. Consequently, only few papers have reported methane emission estimates based on sampling in late winter/during ice melting, further including both bubbling and diffusion of methane.

The results presented are interesting widely to scientific community. Methane is a strong greenhouse gas and updated estimates are needed in order to improve the reliability of emission estimates.

The validity of the approach, quality of the data and quality of presentation including supplementary information are good. Further, the reporting of data and methodology are sufficiently detailed and transparent to enable reproducing the results.

Use of statistics and treatment of uncertainties, the appropriateness of statistical tests, and the accuracy of the description of error bars and probability values are satisfactory.

The conclusions and data interpretation are robust, valid and reliable enough. The reliability of methane emission estimates will be improved when more experimental measurements have been gathered during winter and spring ice melt period.

Response: We appreciate your acknowledgment about the challenges in the field sampling and your positive evaluation of the data quality, methodology, and statistical analysis. We fully agree that future studies with more experimental measurements in winter and spring will enhance the reliability of methane emission estimates.

Extrapolation of results is satisfactory. Previous literature has been mostly appropriately cited, beyond US and Sweden, some Canadian and Finnish studies might be useful. The future scenarios include both predictions for future expansion of thermokarst lake area and ice duration.

Well written manuscript including interesting results – it is editorial decision whether methane is so important compared to other topics. The group has long term experience on this issue and

has published also in Nature Communications.

Response: Relevant citations have been added in the revised manuscript.

Thompson, H. A., White, J. R., Pratt, L. M. & Sauer, P. E. Spatial variation in flux, $\delta^{13}\text{C}$ and $\delta^2\text{H}$ of methane in a small Arctic lake with fringing wetland in western Greenland. Biogeochemistry 131, 17-33, doi:10.1007/s10533-016-0261-1 (2016).

Rinta, P. et al. An inter-regional assessment of concentrations and $\delta^{13}\text{C}$ values of methane and dissolved inorganic carbon in small European lakes. Aquatic Sciences 77, 667-680, doi:10.1007/s00027-015-0410-y (2015).

Walter, K. M., Chanton, J. P., Chapin, F. S., III, Schuur, E. A. G. & Zimov, S. A. Methane production and bubble emissions from arctic lakes: Isotopic implications for source pathways and ages. Journal of Geophysical Research-Biogeosciences 113, doi:10.1029/2007jg000569 (2008).

Walter, K. M., Zimov, S., Chanton, J. P., Verbyla, D. & Chapin III, F. S. Methane bubbling from Siberian thaw lakes as a positive feedback to climate warming. Nature 443, 71-75 (2006).

Phelps, A. R., Peterson, K. M. & Jeffries, M. O. Methane efflux from high-latitude lakes during spring ice melt. Journal of Geophysical Research: Atmospheres 103, 29029-29036 (1998).

Wik, M., Varner, R. K., Anthony, K. W., MacIntyre, S. & Bastviken, D. Climate-sensitive northern lakes and ponds are critical components of methane release. Nature Geoscience 9, 99-105 (2016).

Reviewer #2 (Remarks to the Author):

The manuscript titled “Methane emissions from thermokarst lakes must emphasize the ice-melting impact” reveals an enormous work done to expand our knowledge on remote ecosystems and their often-neglected impact on the global processes. The manuscript is based on 3 types of data – 1. Fieldwork data collected in harsh conditions, in remote areas during ice-out season, 2. Compiled data from previously available studies, and 3. Modelling based on future climatic scenarios. The combination of all three data sets is a great strength of this manuscript, and I would like to congratulate the authors with the all the hard work done to achieve these results. I, however, have some concerns about the presentation of the data, especially since articles published in Nature journals are directed to the wide public and not only one specific group of specialists. While the methodology presented by the authors seems sound and valid, with all the limitations we all face in any field of science, I felt that the presentation of the results and, especially, the discussion of what they tell us could be ameliorated before presenting to the wide auditorium of the journal; hence, I would recommend a major review of the manuscript.

Response: We sincerely appreciate your great comments for our study. The detailed comments are all valuable and very helpful for improving our paper. We have made careful revisions in the discussion and hope our endeavor has fully addressed your comments.

My main concern is that the authors present a wide range of data, but it often feels disconnected from each other. With the way the text is presented, I was sometimes confounded as of the need of certain measurements or analyses – when reading the manuscript, I was periodically led to doubting of the need and meaning of presented results. The messages for the reader could be done more carefully, with clear intent, and a continuous storyline. Below I provide my comments in more detail, but most of them deal with this major issue. I would like to underline that I consider this study of great value, and all the results individually have their importance. The manuscript would significantly increase its value and accessibility if the authors were to carefully reconsider their discussions and try to weave one but complete story.

Response: According to the valuable suggestions, we improved the discussions in the revised version according to the following specific comments. The detailed responses are shown as the following comments.

Furthermore, the manuscript presents a study of regional importance - although Tibetan Plateau is a significant high-altitude permafrost area, and the global scientific community will be interested in learning about such rare findings from this region, it must be clearly acknowledged throughout the manuscript that we are dealing with the results from Tibetan Plateau. The authors did precise this in many instances, but there are whole paragraphs when it is unclear whether we are dealing with interpretations for the thermokarst lakes in general, or only those studied in Tibet.

Response: We have added the Tibetan Plateau in the revised manuscript. The specific revisions are shown as follows:

Lines 114-116: *“To explore the difference in dissolved CH₄ concentrations between ice-free and ice-covered periods on the Tibetan Plateau, we conducted field observations in March-April 2023 and integrated our published data from thermokarst lakes during 2019–2023¹⁸.”*

Lines 138-140: *“It was shown that DOC in the thermokarst lakes on the Tibetan Plateau is closely related to vegetation types¹⁸.”*

Lines 142-147: *“Furthermore, to elucidate the pathways of CH₄ production in*

thermokarst lakes during ice-covered periods on the Tibetan Plateau, we calculated the carbon fractionation factor (α_c) using $\delta^{13}\text{C}$ values of dissolved CO_2 and CH_4 . The α_c values larger than 1.055 indicate CH_4 origin predominantly via CO_2 reduction, while values between 1.040 and 1.055 suggest acetate fermentation^{29,30}. α_c values below 1.040 are likely associated with CH_4 oxidation³¹ (see Methods)."

Lines 147-150: *"Results show that the observed $\delta^{13}\text{C}$ - CH_4 in the water body under lake ice has a median value of -57.9‰ (ranging from -89.6‰ to -26.7‰) (Supplementary Fig. 4a), similar to that of diffusion during ice-free periods on the Tibetan Plateau."*

Lines 153-156: *"This pathway aligns with that of CH_4 emissions via diffusion during ice-free periods¹⁸, but contrasts with that via ebullition during ice-free periods, which is predominantly driven by CO_2 reduction on the Tibetan Plateau¹⁹."*

Lines 170-171: *"Our study sheds light on the mechanistic understanding of CH_4 release dynamics in the Tibetan Plateau thermokarst lakes."*

Lines 229-231: *"To estimate the extent of previously underestimated CH_4 release on the Tibetan Plateau, we used a Monte Carlo approach to randomly sample thermokarst lake CH_4 flux for each vegetation type."*

Lines 266-268: *"Therefore, enhancing more observations, especially for changes in lake ice phenology on the Tibetan Plateau, is crucial for a comprehensive understanding of CH_4 emissions from thermokarst lakes."*

Lines 275-278: *"To predict future changes of thermokarst lakes on the Tibetan Plateau, we used the Random Forest (RF) model to assess the susceptibility distribution of thermokarst lakes and calculate the actual area of thermokarst lake based on the lake area density under different vegetation types (Supplementary Table 5) (see Methods)."*

Lines 278-280: *"The results show that thermokarst lakes on the Tibetan Plateau are expected to increase under SSP scenarios."*

Lines 299-300: *"Our findings indicate significant increases in CH_4 emissions from Tibetan Plateau thermokarst lakes even if lake ice loss is not considered"*

Lines 313-316: *"Taken together, future CH_4 emissions from the Tibetan Plateau thermokarst lakes could reach 135.5–143.8 Gg C yr⁻¹ by 2100 (Fig. 3f), and considering the CO_2 -equivalent emissions of CH_4 , this could potentially offset about 14.1% of the carbon sink in alpine grasslands⁵⁰."*

Lines 132-136. I understand the reason the authors started their interpretations this way; however, I feel it is unnecessary, especially since they do not have much to link the methane concentrations with the vegetation itself. I would recommend skipping the comments on the vegetation type and go directly to the gradient of soil water content (from wet meadows to deserts) and the amount of soil organic matter available around the studied ponds, both of which correspond closely to the presented vegetation type gradient. Wet meadows accumulate more undegraded peat, as wet soils are more likely to form persistent anoxic conditions slowing down the biodegradation, and deserts have less organic matter as there are less vegetation. You discuss most of this already in the next lines of this paragraph, and define the ground and organic matter properties of the studied regions in methods, but this information with different landscapes is worthy longer discussion attaching the soil properties (humidity, organic matter, sensitivity to permafrost thaw). To be sure I am not misunderstood, I do not ask to remove the vegetations types from the study, but to treat them as different landscape with different properties in more wholesome ways; meanwhile discussion based on the plant species themselves is not useful the way it is presented right now.

Response: In the revised version, we have treated vegetation types as the landscapes with different soil organic carbon contents. Because we focus on the dissolved CH_4 concentrations in lake water, dissolved organic carbon (DOC) in lake water provides the crucial substrate for

microbial CH₄ production. The DOC is controlled by soil organic carbon around the lakes within the watershed, which is transported into the thermokarst lakes through hydrological processes. Our previous study showed DOC in the thermokarst lake on the Tibetan Plateau is closely related to vegetation types. Thus, we added the Supplementary Fig. 2 and discussed the effect of dissolved organic carbon (DOC) in lake water availability across the different landscapes as follows:

Lines 130-141: “Additionally, a good correlation is observed between dissolved CH₄ and dissolved organic carbon (DOC) concentrations in lake water (Supplementary Fig. 2). This is attributable to that DOC provides the crucial substrate for microbial CH₄ production in the water column^{23,24}. The gradient of DOC concentration in lake water is controlled by soil organic carbon around the lakes within the watersheds, which is transported into the thermokarst lakes through hydrological processes²⁵⁻²⁸. Thus, the highest CH₄ concentrations are found in thermokarst lakes under the vegetation of alpine swamp meadows (ASM) and meadows (AM), followed by alpine steppe (AS) and desert (AD), evident both ice-covered and ice-free periods (Supplementary Fig. 3). It was shown that DOC in the thermokarst lakes on the Tibetan Plateau is closely related to vegetation types¹⁸. The results suggest that large amounts of CH₄ are trapped in ice-covered thermokarst lakes, exhibiting similar patterns to those during ice-free period.”

Revised Supplementary Fig 2. The correlation between dissolved CH₄ and dissolved organic carbon (DOC) concentrations in thermokarst lake water. Samples were collected during the ice-free periods from 2019 to 2023 at Beilu River, Wudao Liang, and Tanggula of the Tibetan Plateau.

Fig. 1g. It is not clear from how many observations per vegetation type these regressions are based on – it seems that sometimes there are only two points, or so. If the number of observations is that small, I do not consider it worthy to be placed in one of the main figures. This kind of information may be considered as a curiosity (maybe as a supplementary figure?) but not much can be said from it. If I am mistaken, please provide more information, or add all the data points into the graph to make it more informative.

Response: We have updated the figure including all monitoring data points for each vegetation

type. Additionally, we have refined the presentation of figure to make it more informative and visually clear, for example changing the solid circles to hollow circles. The revised Figure 1 is shown as follows:

Revised Fig. 1 | Field observations of thermokarst lakes during ice-covered period on the Tibetan Plateau. (a) Distribution of the 56 monitored thermokarst lakes during ice-covered period (this study) and 162 thermokarst lakes during ice-free period¹⁸. A total of 409 field observations were conducted from June 2019 to April 2023. Permafrost and vegetation distribution data are sourced from existing distribution dataset^{12,21}. (b-e) The images depict field observations during the ice-covered periods from March to April, captured by Pengsi Lei and Mei Mu. (f) Density frequency of dissolved CH₄ concentrations and their comparison between the ice-covered period and ice-free period. The lines indicate CH₄ concentrations of individual samples during ice-free and ice-covered periods. (g) The relationships of dissolved CH₄ concentrations between the ice-covered and ice-free periods under vegetation types of alpine swamp meadow (ASM), alpine meadow (AM), alpine steppe (AS) and alpine desert (AD). Notably, the circles represent the individual lakes observed during both the ice-free and ice-covered periods. The corresponding values indicate the average CH₄ concentration within the same lake during the two periods. The shaded area represents the 95% confidence interval. * $P < 0.05$, ** $P < 0.01$, *** $P < 0.001$.

Paragraph starting with the line 142. Isotopic data may often add important insights into methane-related studies; however, in this case I do not see your data as adding any new value to serve your objectives. The authors may want to consider rewriting this paragraph to make it less descriptive-only and add more interpretations, and comparisons with other studies done in winter and summer time in Finnish boreal lakes, or Canadian Arctic lakes. What do these isotopic values tell us about the present and future methane emissions, methane sources? How does the fact that ebullition from these thermokarst lakes come from CO₂ reduction affect our understanding of future predictions? Do we expect these values to change with time?

Response: We have rewritten this paragraph carefully and hope this clarification address your concerns. Firstly, we clearly showed the $\delta^{13}\text{C-CH}_4$ in water body under thermokarst lake ice and implied the pathway for CH₄ production during ice-covered periods on the QTP. Then, the difference in pathway of CH₄ production between ebullition and diffusion is clarified by adding more interpretations. Secondly, the finding is compared with that in Arctic lakes, which further deepens the mechanistic understanding of CH₄ release dynamics in the Tibetan Plateau thermokarst lakes. Finally, we added the discussion about the effect of future climate warming on CH₄ production and emissions. The revised contents are shown as follows:

Lines 142-171: *“Furthermore, to elucidate the pathways of CH₄ production in thermokarst lakes during ice-covered periods on the Tibetan Plateau, we calculated the carbon fractionation factor (α_c) using $\delta^{13}\text{C}$ values of dissolved CO₂ and CH₄. The α_c values larger than 1.055 indicate CH₄ origin predominantly via CO₂ reduction, while values between 1.040 and 1.055 suggest acetate fermentation^{29,30}. The α_c values below 1.040 are likely associated with CH₄ oxidation³¹ (see Methods). Results show that the observed $\delta^{13}\text{C-CH}_4$ in the water body under lake ice has a median value of -57.9‰ (ranging from -89.6‰ to -26.7‰) (Supplementary Fig. 4a), similar to that of diffusion during ice-free periods on the Tibetan Plateau. The median α_c value of water storage during ice-covered periods is 1.038 (n=140) (Supplementary Fig. 4b), which indicates that acetate fermentation is the primary pathway for CH₄ production during ice-covered periods, accompanied by significant oxidation^{29,31-33}. This pathway aligns with that of CH₄ emissions via diffusion during ice-free periods¹⁸, but contrasts with that via ebullition during ice-free periods, which is predominantly driven by CO₂ reduction on the Tibetan Plateau¹⁹. Similarly, high-emission point sources and hotspots in Siberian lakes are primarily driven by CO₂ reduction, while lower-emission processes are influenced by acetate fermentation²⁹. However, in Western Greenland, it was shown that the primary production pathway for CH₄ released through ebullition is acetate fermentation³⁴, whereas in Finland, CH₄ emissions via diffusion are mainly driven by CO₂ reduction³⁵. The discrepancy is attributed to that the different composition of microbial communities in aquatic systems can control the CH₄ production pathways^{36,37}. Additionally, the environmental factors, such as temperature³⁸, salinity¹⁹, and substrates available for methanogenesis³⁹, can further influence the pathway of CH₄ production.*

Under future climate scenarios, the extension of ice-free periods and warmer lake water are expected to enhance the proportion of CH₄ ebullitive emissions on the Tibetan Plateau. Particularly, changes in lake water temperature and dissolved oxygen content can shift the methanogenic pathway, with more CH₄ is likely released through ebullition driven by CO₂ reduction³⁸, potentially increasing the total CH₄ emissions from thermokarst lakes. Our study sheds light on the mechanistic understanding of CH₄ release dynamics in the Tibetan Plateau thermokarst lakes.”

Line 164. “...we quantified the fluxes through diffusion and ebullition during ice-164 free period...” “I do not consider it fitting to declare that the authors quantified the fluxes, when all the ebullition data in this study, if I understood well, comes from other previously published

works. It is ok to use the other data, but in that case the discussion needs to be appropriately constructed. The strength of this study is the data from the ice-out period, which should be put forward, and only then compared to the open-water data available in previous studies. I consider it sufficient to mention the ice-free period emissions as you do with the beginning of the next paragraph (line 180), it could be your first mention there.

Response: To avoid any confusion, we have revised this sentence as follows:

Lines 174-177: *“To systematically evaluate the total annual CH₄ release from thermokarst lakes on the Tibetan Plateau, we quantified the water storage and ice bubble storage by conducting the field observations during ice-cover periods and synthesizing our published fluxes during ice-free period¹⁸ (see Methods).”*

Line 171. “We found that CH₄ release from thermokarst lakes was mainly from water storage during ice-melting”. The structure of the sentence leads to diverging interpretations – do the authors mean that the diffusive emission from water storage is the main pathway of methane evasion, when considering only the ice melt period? Or is it rather considering the whole annual emissions? Please rephrase to be more precise.

Response: We have clarified this expression as follows:

Lines 179-180: *“The result shows that during the ice-melting period, CH₄ release from thermokarst lakes primarily originates from water storage (Fig. 2a).”*

Lines 253-266. Any reader appreciates honest presentation of study limitations, and as someone working in high Arctic lakes, I understand very well the constraints. I do not know, however, if this section needs so many excuses – these constraints are normal and expected. It would be enough to briefly mention that harsh and remote fieldwork conditions limit how spatially and temporally representative your data is. I would avoid calling them “unexpected uncertainties”, though, as anyone could expect uncertainties from a couple of samples for such vast areas the study represents. To add to your list of uncertainties, I would like to see the acknowledgement of limitation from sampling depth perspective – if I understood well, you only took one sample from under ice, neglecting vertical differences in water column.

Response: In the revised version, we have deleted the discussion about the limitation of harsh and remote fieldwork conditions, and added the uncertainties due to the lack of vertical differences in lake water column. It was shown as follows:

Lines 260-268: *“Despite overcoming challenges such as harsh climatic conditions during the ice-covered period and limited accessibility in high-altitude permafrost regions, our estimation has possible uncertainties due to field observations. On the one hand, the sampling focuses on a specific depth of the water body beneath lake ice, resulting in the vertical difference in CH₄ concentrations is neglected. On the other hand, the duration of ice-covered and ice-melting periods was estimated using remote sensing⁴³, which might not represent all small thermokarst lakes. Therefore, enhancing more observations, especially for changes in lake ice phenology on the Tibetan Plateau, is crucial for a comprehensive understanding of CH₄ emissions from thermokarst lakes.”*

Paragraph starting with line 274. The first half of this paragraph describes the methodology (some details are fine, I imagine, but now it seems to be the main focus, in the result-discussion section). Then the authors move on to saying that their results are comparable to already existing observations. I do not wish to reduce the importance of modeling and the work the authors did to achieve these results, but in order for this whole section to be relevant, it needs to be presented differently. Maybe instead a couple of sentences within the sections on future

emissions, or the perspectives?

Response: We have rewritten the discussion by removing repetitive methodological descriptions. The methodological details (e.g., machine learning approaches and susceptibility analysis) are primarily presented in the Methods section and the Supplementary Information. In the revised version, the Results and Discussion sections focus on the findings, particularly their connection to future emissions and broader implications for alpine thermokarst lakes under a warming climate. We hope that this revised presentation clarifies and strengthens our main conclusions.

Lines 275-294: *“To predict future changes of thermokarst lakes on the Tibetan Plateau, we used the Random Forest (RF) model to assess the susceptibility distribution of thermokarst lakes and calculate the actual area of thermokarst lake based on the lake area density under different vegetation types (Supplementary Table 5) (see Methods). The results show that thermokarst lakes on the Tibetan Plateau are expected to increase under SSP scenarios. By 2100, the total area of thermokarst lakes is projected to reach 3,912 km² (increase by 85.9% compared with 2020) under SSP1-2.6, 3,926 km² (86.5%) under SSP2-4.5, and 4,102 km² (94.9%) under SSP5-8.5 (Supplementary Fig. 8). These changes correspond with the previous remote sensing observations that showing an increase of 14.7% per-decade between 1969 and 2010 and 20.0% per-decade between 2010 and 2019⁵¹. Our results forecast thermokarst lakes area will increase by 33.2% per decade between 2020 to 2040, which is higher than the previous increasing rate due to future warming and wetting in the alpine region (Supplementary Fig. 9). Although future prediction of thermokarst lakes relies more on the process model and need more verification, abrupt permafrost thaw cannot be simulated by Earth System Models (ESMs). Our results show the response of these lakes on the Tibetan Plateau to future climate change through considering the influencing factors of climate, topography, hydrology, soil, permafrost, and human activity (Supplementary Table 3). Our study highlights the future changes of alpine thermokarst lakes and their ecological and environmental impacts.”*

Lines 629-645: *“We analyzed a total of 161,341 thermokarst lakes¹⁵. Prior to machine learning modeling, we used the ENMTools to filter out redundant samples, ensuring that only one lake was present in each 0.0083° × 0.0083° grid. Ultimately, we selected 71,164 thermokarst lakes as training samples for the model. Meanwhile, we selected nine environmental factors influencing the distribution of thermokarst lakes. These factors include topography, hydrology, soil, human activities, permafrost and climate (Supplementary Table 3). The topography factors, including elevation and slope, was derived from high-precision Global Digital Elevation Model (DEM) (ETOPO Global Relief Model, National Centers for Environmental Information (NCEI) (noaa.gov)). The elevation data were directly obtained from the DEM, while the slope was calculated based on the DEM. The Topographic Wetness Index (TWI) is calculated using DEM⁸⁹. Soil factors, including sand and silt, were obtained from the Harmonized World Soil Database version 2.0 (<https://doi.org/10.4060/cc3823en>). The human footprint was from Last of the Wild, v2⁹⁰. To simulate the current susceptibility distribution of thermokarst lakes, climatic factors, including monthly maximum air temperature and precipitation, were obtained from WorldClim. The permafrost factor of active layer thickness (ALT) was sourced from a published dataset⁹¹.”*

Lines 647-664: *“To predict future changes in thermokarst lake susceptibility under SSP1-2.6, SSP2-4.5, SSP3-7.0, and SSP5-8.5, we used monthly air temperature, precipitation (<https://www.worldclim.org/>), and ALT⁹² data from five general circulation models (GCMs). We refined the resolution to 1 km using the delta approach. All data were masked to the study area and resampled to a spatial resolution of 0.0083° × 0.0083°. Our study used six machine*

learning models to assess the susceptibility of thermokarst lakes: Random Forest (RF), Generalized Additive Model (GAM), Generalized Boosted Regression Model (GBM), Classification and Regression Tree Analysis (CTA), Artificial Neural Network (ANN), and MaxEnt. We evaluated the performance of models using Receiver Operating Characteristic (ROC) curves, Kappa statistics, and true skill statistics (TSS) (Supplementary Table 4). Based on these evaluations, we select RF model with the best performance for modeling. We set thresholds at 0.2, 0.4, 0.6, and 0.8 to classify susceptibility into five levels: very low, low, medium, high, and very high⁹³. We identified the regions with moderate to very high levels as thermokarst lake affected areas. Then we estimated the lake area density (i.e., the area of lakes within a pixel) to calculate the future actual area of thermokarst lakes under different vegetation types. The standard error of the area density was used to quantify the uncertainty in these future projections (Supplementary Table 5).”

Section on future methane emissions (from line 301). While the title of the section is promising, I found it rather dry (lots of statistics and future projections in percentages of increase; information that can be presented in a table instead of two paragraphs) and sometimes confusing. I do not wish to discourage the authors; however, the presented results are interesting and useful, but the discussion of their implications was often missing. I would recommend to discuss aspects like: 1. what the shorter period of ice cover would mean for future methane winter storage and emissions – more lake expansion?, less methane accumulation?, more ebullition during longer summers?; 2. How do future prediction vary based on the vegetation types? Do all lakes behave the same? Is it possible to model lake expansion or methane emission based on regional soil/vegetation type?; 3. How do these predictions for Tibetan Plateau differ from other permafrost regions in the world?

Response: We added a new table in order to make more clear information (Supplementary Table 6). Furthermore, we made revisions to improve the discussion according to three aspects: (i) the effect of shorter ice-covered period on future CH₄ in water storage and emissions, (ii) the effect of vegetation types in future prediction, (iii) the difference between the Tibetan Plateau and Arctic thermokarst lakes. The specific revised contents are shown as follows:

Lines 297-333: *“To show the effects of future lake expansion and ice loss on CH₄ release, we projected future CH₄ emissions with or without considering changes in thermokarst lake ice (see methods). Our findings indicate significant increases in CH₄ emissions from Tibetan Plateau thermokarst lakes even if lake ice loss is not considered (Supplementary Table 6). Specifically, CH₄ emissions during the ice-melting period are expected to rise by 76.7%-77.8% by 2050 and 93.8%-105.8% by 2100 under the SSP1-2.6, SSP2-4.5, and SSP5-8.5 (Fig. 3a). During the ice-free period, the emissions could increase by 95.9%-107.93% by 2100 (Fig. 3c), with total annual emissions potentially reaching 116.66–117.34 Gg C yr⁻¹ by 2050 and 128.00–135.85 Gg C yr⁻¹ by 2100 (Fig. 3e). However, over the past 40 years, the average ice-covered duration on the Tibetan Plateau has decreased at a rate of 0.2 days per year (Supplementary Fig. 10)⁴³. Taking the loss of lake ice into account, by 2100, a shorter ice-covered duration could reduce the increase in CH₄ emissions during ice-melting periods but enhance it during ice-free periods (Fig. 3b, d). This is because the shortened ice-covered period means less CH₄ accumulation in the water beneath the ice^{52,53} and more ebullition during ice-free periods⁵⁴. Moreover, longer ice-free period can boost methanogenic activity, further enhancing CH₄ emissions^{55,56}. Taken together, future CH₄ emissions from the Tibetan Plateau thermokarst lakes could reach 135.5–143.8 Gg C yr⁻¹ (Fig. 3f), and considering the CO₂-equivalent emissions of CH₄, this could potentially offset about 14.1% of the carbon sink in alpine grasslands⁵⁰.”*

The future increase of CH₄ emissions on the Tibetan Plateau is possibly different with the Arctic lakes⁷. This is attributable to the different changes of thermokarst lake areas between

the Arctic and Tibetan Plateau, specifically the Arctic is undergoing the extensive draining of thermokarst lakes^{57,58}, whereas thermokarst lakes on the Tibetan Plateau are expanding rapidly in response to climate change^{16,51,59,60}. To date, the estimate of CH₄ release rarely involves the changes in areas and ice phenology of thermokarst lakes. The advantage of this study is that we fully consider the changes of thermokarst lake based on the remote sensing monitoring and future climate scenarios. However, our simulations did not fully incorporate dynamic vegetation changes. The variations of vegetation and permafrost stability highlight that not all thermokarst lakes respond uniformly, underscoring the need for more detailed vegetation change data to improve the projections, particularly under warming-driven shifts in vegetation types. Future increase in CH₄ emissions from thermokarst lakes on the Tibetan Plateau highlights the growing importance in global greenhouse gas inventory data. Our study demonstrates that the expansion of thermokarst lakes and shortening of lake ice-covered duration accelerate CH₄ emissions, playing a crucial role in predicting permafrost carbon feedback to climate change.”

Supplementary Table 6. Projected CH₄ emissions from thermokarst lakes on the Tibetan Plateau under SSP scenarios during ice-free and ice-melting periods by 2050 and 2100. Data are presented as mean ± standard error (SE).

Periods		CH ₄ emissions (Gg C yr ⁻¹)			
		Ice-melting period		Ice-free period	
		No lake ice change	Lake ice loss	No lake ice change	Lake ice loss
1969	Luo et al., 2022 ²	5.93	6.39	28.71	26.96
2020	This study	11.21 ± 1.64		54.24 ± 9.90	
	SSP1-2.6	19.83 ± 5.17	18.92 ± 4.93	96.99 ± 23.44	100.46 ± 24.28
2050	SSP2-4.5	19.93 ± 5.19	19.01 ± 4.95	97.42 ± 23.55	100.90 ± 24.39
	SSP5-8.5	19.80 ± 5.16	18.89 ± 4.93	96.86 ± 23.41	100.32 ± 24.25
	SSP1-2.6	21.72 ± 5.65	19.06 ± 4.95	106.27 ± 25.74	116.41 ± 28.20
2100	SSP2-4.5	21.88 ± 5.69	19.20 ± 4.99	107.04 ± 25.93	117.25 ± 28.40
	SSP5-8.5	23.06 ± 5.99	20.23 ± 5.25	112.79 ± 27.34	123.55 ± 29.95

REVIEWERS' COMMENTS

Response to Reviewers of NCOMMS-24-50735A

Reviewer #2 (Remarks to the Author):

Once again, I would like to congratulate the authors for their hard work in collecting and processing the data and writing this manuscript. My concerns, expressed in the previous round of reviews, have been answered, and I would generally approve the publication of the manuscript with minor recommendations expressed below:

1. When reading the limited publications that touch winter methane story in ice-covered lakes, it is not unusual to see that methane concentrations decrease through winter, assumingly, due to the combination of slower rates of methanogeny (lower temperatures, less labile organic matter) and continuous work of methanotrophs. In the lakes you sampled, it seems to not be the case; nevertheless, it would further improve your discussion if you addressed this divergence in methane (vs CO₂) accumulation under ice. Do you have any suggestions, why lakes in Tibet continue producing CH₄ throughout winter? Can this be answered comparing isotopic data between your and other winter studies? How would such divergence in winter methane stories affect the future models and their interpretations?

Response: Thank you for your valuable review. According to the great comments, we have improved the discussion in the revised version as follows:

Lines 125-127: *"It is attributable to that methanogenesis continues under the lake ice due to the anaerobic environments and barrier effect of lake ice^{9,22}, leading to the accumulation and entrapment of CH₄ in ice-covered thermokarst lakes."*

Lines 155-157: *"This finding further implies that CH₄ production continues throughout the winter in the Tibetan Plateau thermokarst lakes."*

Lines 174-179: *"Although regional divergence exists in the CH₄ accumulation during ice-covered periods³⁹⁻⁴¹, potentially affecting future CH₄ emission prediction, the results shed light on the mechanistic understanding of CH₄ release dynamics in the Tibetan Plateau thermokarst lakes. Our study highlights the need to account for winter CH₄ production in thermokarst lakes, which may substantially contribute to future annual CH₄ emissions under ongoing climate warming."*

2. Lines 130-141. ..."DOC provides the crucial substrate for microbial CH₄ production in the water column"... from my experience, it is rather unusual (however, not unbelievable) that methane correlates with DOC... as methanogeny mostly happens in the sediments (it has been proven that some of it may occur in water column as well; but for shallow lakes I would still lean towards blaming the sediments) DOC and CH₄ concentrations could correlate but not necessary due to direct effects. Methanogens are often presented as sensitive to organic matter quality. I would recommend checking more studies regarding the topic (I can see now that you cite two studies from the same regions, one of which is based on rivers). In short, while DOC and CH₄ correlation is not surprising, I would recommend using a different approach when explaining it.

Response: According to your great comment, we analyzed the relationship between dissolved CH₄ concentrations and sediment organic carbon contents in the thermokarst lakes. We updated the Supplementary Figure 2 and revised the discussion as follows:

Lines 130-141: *“Additionally, dissolved CH₄ concentrations are closely related to sediment organic carbon contents in thermokarst lakes with different vegetation types on the Tibetan Plateau (Supplementary Fig. 2). This is attributable to that methanogenesis primarily occurs in the lake sediments, where organic carbon provides crucial substrate for microbial CH₄ production²³⁻²⁵. The gradient of sediment organic carbon contents in thermokarst lakes is controlled by soil organic carbon around the lakes within the watersheds, which is transported into the thermokarst lakes through hydrological processes^{26,27}. Thus, the highest CH₄ concentrations are found in thermokarst lakes under the vegetation of alpine swamp meadows (ASM) and meadows (AM), followed by alpine steppe (AS) and desert (AD), evident both ice-covered and ice-free periods (Supplementary Fig. 3). It was shown that sediment organic carbon contents in the thermokarst lakes is closely related to vegetation types on the Tibetan Plateau^{18,26}.”*

3. A general recommendation to re-evaluate the main title of the manuscript. While it somewhat fits your study design, I think you could make it less limited (it sounds as a paper describing recommendations for future methodologies in the field; and your implication might go beyond the ice-melting period, even if it represents "only" 17% of annual emissions), but also less generic (you studied thermokarst lakes on Tibetan Plateau, and as we have already discussed, the results do not represent thermokarst lakes globally).

Response: Yes, the title has been changed into “Methane emissions from thermokarst lakes must emphasize the ice-melting impact on the Tibetan Plateau”.